



# A mass-weighted atmospheric isentropic coordinate for mapping chemical tracers and computing inventories

Yuming Jin[1], Ralph F. Keeling[1], Eric J. Morgan[1], Eric Ray[2], Nicholas C. Parazoo[3], Britton B. Stephens[4]

[1]Scripps Institution of Oceanography, University of California San Diego, La Jolla, CA 92093, USA

[2]National Oceanic and Atmospheric Administration, Boulder, CO 80305, USA

[3]Jet Propulsion Laboratory, California Institute of Technology, Pasadena, CA 91109, USA

[4]National Center for Atmospheric Research, Boulder, CO 80301, USA

*Correspondence to*: Yuming Jin (y2jin@ucsd.edu)

**Abstract.** We introduce a transformed isentropic coordinate $M_{\theta e}$, defined as the dry air mass under a given equivalent potential
temperature surface ($\theta_e$) within a hemisphere. Like $\theta_e$, the coordinate $M_{\theta e}$ follows the synoptic distortions of the atmosphere,
but unlike $\theta_e$, has a nearly fixed relationship with latitude and altitude over the seasonal cycle. Calculation of $M_{\theta e}$ is
straightforward from meteorological fields. Using observations from the recent HIPPO and Atom airborne campaigns, we map
the $CO_2$ seasonal cycle as a function of pressure and $M_{\theta e}$, where $M_{\theta e}$ is thereby effectively used as an alternative to latitude. We
show that the $CO_2$ cycles are more constant as a function of pressure using $M_{\theta e}$ as the horizontal coordinate compared to
latitude. Furthermore, short-term variability of $CO_2$ relative to the mean seasonal cycle is also smaller when the data are
organized by $M_{\theta e}$ and pressure than when organized by latitude and pressure. We also present a method using $M_{\theta e}$ to compute
mass-weighted averages of $CO_2$ on a hemispheric scale. Using this method with the same airborne data and applying
corrections for limited coverage, we resolve the average $CO_2$ seasonal cycle in the Northern Hemisphere (mass weighted
tropospheric climatological average for 2009-2018), yielding an amplitude of 7.8 ± 0.14 ppm and a downward zero-crossing
at Julian day 173 ± 6.1 (i.e., late June). $M_{\theta e}$ may be similarly useful for mapping the distribution and computing inventories of
any long-lived chemical tracer.

## 1 Introduction

The spatial and temporal distribution of long-lived chemical tracers like $CO_2$, $CH_4$, and $O_2/N_2$ typically includes regular
seasonal cycles and gradients with latitude (Conway and Tans, 1999; Ehhalt, 1978; Randerson et al., 1997; Rasmussen and
Khalil, 1981; Tohjima et al., 2012). These patterns are evident in climatological averages but are potentially distorted on short
time scales by synoptic weather disturbances, especially at middle to high latitudes (i.e. poleward of 30° N/S) (Parazoo et al.,
2008; Wang et al., 2007). With a temporally-dense dataset such as from satellite remote sensing or tower in-situ measurements,



climatological averages can be created by averaging over this variability. For temporally sparse datasets such as from airborne campaigns, it may be necessary to correct for synoptic distortion.

One method for reducing the impact of synoptic variability is to evaluate tracer data on isentropic coordinates, i.e. based on potential temperature $\theta$ (Hess, 2005; Miyazaki et al., 2008; Parazoo et al., 2011, 2012). As air parcels move with synoptic disturbances, $\theta$ and the tracer tend to be similarly displaced so that the $\theta$-tracer relationship is relatively conserved (Keppel-Aleks et al., 2011). Furthermore, vertical mixing tends to be rapid on $\theta$ surfaces, so $\theta$ and tracer contours are often nearly parallel (Barnes et al., 2016). On the other hand, $\theta$ varies greatly with latitude and altitude over seasons due to changes in

heating and cooling with solar insolation, which complicates the interpretation of $\theta$-tracer relationships on seasonal time scales.

During analysis of airborne data from the HIAPER Pole-to-Pole Observations (HIPPO) (Wofsy, 2011) and the Atmospheric Tomography Mission (ATom) (Prather et al., 2018) airborne campaigns, we have found it useful to transform potential temperature into a mass-based unit, $M_\theta$, which we define as the total mass of dry air under a given isentropic surface in the hemisphere. In contrast to $\theta$, which has large seasonal variation, $M_\theta$ has a more stable relationship to latitude and elevation,

while varying in parallel with $\theta$ on synoptic scales. Also, for a tracer which is well-mixed on $\theta$, a plot of this tracer versus $M_\theta$ can be directly integrated to yield the inventory of the tracer, because $M_\theta$ directly corresponds to the mass of air. We note that a similar concept to $M_{\theta e}$ has been introduced in the stratosphere by Linz et al. (2016).

Several choices need to be made in the definition of $M_\theta$, including defining boundary conditions (e.g. in altitude and latitude) for mass integration and whether to use potential temperature $\theta$ or equivalent potential temperature $\theta_e$. Here, for boundaries,

we use the dynamical tropopause (based on potential vorticity unit, PVU) and the Equator, thus integrating the dry air mass of the troposphere in each hemisphere. We also focus on $M_\theta$ defined using equivalent potential temperature ($\theta_e$) to conserve moist static energy in the presence of latent heating during vertical motion, which improves alignment between mass transport and mixing especially within storm tracks in mid-latitudes (Parazoo et al., 2011; Pauluis et al., 2008, 2010). We call this tracer $M_{\theta e}$.

In this paper we describe the method for calculating $M_{\theta e}$ and discuss its variability on synoptic to seasonal scales. We also discuss the time variation of the $\theta_e$-$M_{\theta e}$ relationship within each hemisphere and explore the stability of $M_{\theta e}$ and $\theta_e$-$M_{\theta e}$ relationship using different reanalysis products. To illustrate the application of $M_{\theta e}$, we map $CO_2$ data from two recent airborne campaigns (HIPPO and ATom) on $M_{\theta e}$. Further, we show how $M_{\theta e}$ can be used to accurately compute the average $CO_2$ concentration over the entire troposphere of the Northern Hemisphere using measurements from the same airborne campaigns.

We examine the accuracy of this method and propose an appropriate way to sample the atmosphere with aircraft to compute the average of a chemical tracer within a large zonal domain.





## 2 Methods

### 2.1 Meteorological reanalysis products

The calculation of $M_{\theta e}$ requires the distribution of dry air mass and $\theta_e$. For these quantities, we alternately use three reanalysis

products: ERA-Interim (Dee et al., 2011), NCEP2 (Kanamitsu et al., 2002), and Modern-Era Retrospective analysis for Research and Applications Version 2 (MERRA-2) (Gelaro et al., 2017). These products have the following resolution in latitude, longitude, vertical level count, and time: ERA-Interim (2.5°, 2.5°, 32, 6-hourly), NCEP2 (2.5°, 2.5°, 17, daily), MERRA-2 (2.5°, 2.5°, 42, 3-hourly). For ERA-Interim and MERRA-2, we average the 6-hourly or 3-hourly fields to yield daily fields.

### 2.2 Equivalent potential temperature ($\theta_e$) and dry air mass (M) of the atmospheric fields

We compute $\theta_e$ (K) using the following expression:

$$\theta_e = \left( T + \frac{L_v(T)}{C_{pd}} \cdot w \right) \cdot \left( \frac{P_0}{P} \right)^{\frac{R_d}{C_{pd}}} \tag{1}$$

from Stull (2012). T (K) is the temperature of air, w (kg water vapor per kg air mass) is the water vapor mixing ratio, $R_d$ (287.04, J kg$^{-1}$ K$^{-1}$) is the gas constant for air, $C_{pd}$ (1005.7 J kg$^{-1}$ K$^{-1}$) is the specific heat of dry air at constant pressure, $P_0$

(1013.25, mbar) is the reference pressure at the surface, and $L_v(T)$ is the latent heat of evaporation at temperature T. $L_v(T)$ is defined as 2406 kJ kg$^{-1}$ at 40 °C, and 2501 kJ kg$^{-1}$ at 0 °C and scales linearly with temperature.

We compute water vapor mixing ratio (w) from relative humidity provided by the reanalysis products and the saturation mixing ratio of water vapor following Bolton (1980).

We compute the total air mass of each grid cell x at time t, $M_x(t)$, from the product of pressure range and surface area, and

divided by a latitude and height dependent gravity constant provided by Arora et al. (2011). The dry air mass is then computed by subtracting the water mass, computed from relative humidity, saturation water vapor mass mixing ratio, and total air mass of the grid cell.

Since this study focuses on tracer distributions in the troposphere, we compute $M_{\theta e}$ with an upper boundary at the dynamical tropopause defined as the 2 PVU (potential vorticity units, $10^{-6}$ K kg$^{-1}$ m$^2$ s$^{-1}$) surface. As PV is not a standard output product

in NCEP2, we linearly interpolate PV from ERA-Interim to NCEP2 on its pressure coordinate.

ERA-Interim and NCEP2 include hypothetical levels below the true land/sea surface which we exclude in the calculation of $M_{\theta e}$.

Further details are included in Supplementary S1.



## 2.3 Determination of $M_{\theta e}$

We show a schematic of the conceptual basis for the calculation of $M_{\theta e}$ in Figure 1. To compute $M_{\theta e}$, we sort all tropospheric grid cells in the hemisphere by increasing $\theta_e$, and sum the dry air mass over grid cells following

$$M_{\theta_e}(\theta_e, t) = \sum M_x(t)|_{\theta_{e_x} < \theta_e} \qquad (2)$$

where $M_x(t)$ is the dry air mass of each grid cell x at time t, and $\theta_{e_x}$ is the equivalent potential temperature of the grid cell. The sum is over all grid cells with $\theta_{e_x}$ less than $\theta_e$.

This calculation yields a unique value of $M_{\theta e}$ for each value of $\theta_e$. We refer to the relationship between $\theta_e$ and $M_{\theta e}$ as the "$\theta_e$-$M_{\theta e}$ look-up table", which we generate at daily resolution. We provide this look-up table for each hemisphere computed from ERA-Interim from 1980 to 2018 with daily resolution and 1 K interval (see data availability).

## 3 Characteristics of $M_{\theta e}$

### 3.1 Spatial and temporal distribution of $M_{\theta e}$

Figure 2 shows snapshots of the distribution of zonal average $\theta_e$ and $M_{\theta e}$ with latitude and pressure at two time slices (1 January 2009, 1 July 2009). $M_{\theta e}$ is not continuous across the Equator because it is defined separately in each hemisphere. By definition, $M_{\theta e}$ surfaces are always exactly parallel to $\theta_e$ surfaces, which decrease with latitude and generally increase with altitude. Whereas, the $\theta_e$ surfaces vary by up to 20 degrees in latitude over seasons, the meridional displacement of $M_{\theta e}$ is much smaller, as expected, because the displacement of atmospheric mass over seasons is small. Particularly in latitudes of the Hadley

circulation (Equatorward of 30° N/S) and in summer, $M_{\theta e}$ and $\theta_e$ both exhibit strong secondary maxima at the surface, driven by high water vapor. From the contours in Figure 2, this surface branch of high $M_{\theta e}$ and $\theta_e$ appears disconnected from the upper tropospheric branch. In fact, we expect these two branches are connected through air columns undergoing deep convection, which are not resolved in the zonal means shown in Figure 2. The existence of these two branches limits some applications of $M_{\theta e}$, as discussed in Section 4.

Figure 3 shows the zonal average meridional displacement of $\theta_e$ and $M_{\theta e}$ with daily resolution. In summer, $M_{\theta e}$ surfaces displace poleward in the lower troposphere but equatorward in the upper troposphere. The displacements in the lower troposphere (925 mbar) are greater in the Northern Hemisphere, where the $M_{\theta e} = 140$ ($10^{16}$ kg) surface, for example, displaces poleward by 10 degrees in latitude between winter and summer (Figure 3b). The seasonal displacement of $M_{\theta e}$ surfaces is closely associated with the seasonality of vertical sloping of $\theta_e$ surfaces (Figure 2). As the mass under each $M_{\theta e}$ surface is

always constant, the change in tilt must cause the meridional displacement. In the summer, the tilt is steeper (due to increased deep convection) so $M_{\theta e}$ surfaces move poleward in the lower troposphere but move equatorward in the upper troposphere. Beside the seasonal variability, Figure 3 also shows evident synoptic-scale variability.


Since the tilting of $\theta_e$ surfaces has an impact on the seasonal displacement of $M_{\theta e}$ surfaces, the contribution of different pressure levels to the mass of a given $M_{\theta e}$ bin must also vary with season. In Figure 4, we show these contributions as two daily snapshots on 1 January 2009 and 1 July 2009. Low $M_{\theta e}$ bins consist of air masses mostly below 500 mbar near the Pole. As $M_{\theta e}$ increases, the contribution from the upper troposphere gradually increases while the contribution from the surface to 800 mbar decreases to its minimum at ~ 100 to 120 ($10^{16}$ kg). The contribution from the surface to 800 mbar increases as $M_{\theta e}$ increases above 120 ($10^{16}$ kg). On the other hand, the contribution of air mass below 800 mbar is always higher in the summer hemisphere at high $M_{\theta e}$ bins.

## 3.2 $\theta_e$-$M_{\theta e}$ relationship

Figure 5 compares the temporal variation of $M_{\theta e}$ of several given $\theta_e$ surfaces (i.e., $\theta_e$-$M_{\theta e}$ look-up table) computed from different reanalysis products for 2009. The deviations are indistinguishable between ERA-Interim and MERRA-2, except near $\theta_e = 340$ K, where MERRA-2 is systematically lower than ERA-Interim by 1.5 to 6.5 ($10^{16}$ kg). NCEP2 shows slightly larger deviations from ERA-Interim, but less than 8.5 ($10^{16}$ kg). The products are highly consistent in seasonal variability, and they also show agreement on synoptic time scales. The small difference between products is expected because of different resolutions and methods (Mooney et al., 2011). We expect these differences would be negligible for most applications of $M_{\theta e}$.

Figure 5 shows that, in both hemispheres, $M_{\theta e}$ reaches its minimum in summer and maximum in winter for a given $\theta_e$ surface, with the largest seasonality at the lowest $\theta_e$ (or $M_{\theta e}$) values. The seasonality decreases as $\theta_e$ increases, following the reduction in the seasonality of shortwave absorption at lower latitudes (Li and Leighton, 1993). The seasonality is smaller in the Southern Hemisphere, consistent with the larger ocean area and hence greater heat capacity and transport (Fasullo and Trenberth, 2008; Foltz and McPhaden, 2006). Figure 5 also shows that $M_{\theta e}$ has significant synoptic-scale variability but smaller than the seasonal variability. Synoptic variability is typically larger in winter than summer, as discussed below.

### 3.3 Relationship to diabatic heating and mass fluxes

A key step of the application of $M_{\theta e}$ for interpreting tracer data is the generation of the look-up table that relates $\theta_e$ and $M_{\theta e}$. In this section, we address a tangential question of what controls the temporal variation of the look-up table, which is not necessary for the application but may be of fundamental meteorological interest.

As shown in Appendix A, the temporal variation of the lookup table, $\dot{M}_{\theta_e} = \frac{\partial}{\partial t} M_{\theta_e}(\theta_e, t)$, can be related to underlying mass and heat fluxes according to

$$\dot{M}_{\theta_e} = -\frac{1}{C_{pd}} \frac{\partial Q_{dia}(\theta_e, t)}{\partial \theta_e} + m_T(\theta_e, t) + m_E(\theta_e, t) \tag{3}$$





where $\frac{\partial Q_{dia}(\theta_e, t)}{\partial \theta_e}$ (J s$^{-1}$ K$^{-1}$) is the effective diabatic heating, integrated over the full $\theta_e$ surface per unit width in $\theta_e$, $m_T(\theta_e, t)$ (kg

s$^{-1}$) is the net mass flux across the tropopause and $m_E(\theta_e, t)$ (kg s$^{-1}$) is the net mass flux across the Equator, including all air

with equivalent potential temperature less than $\theta_e$. $Q_{dia}$ has contributions from internal heating without ice formation ($Q'_{int}$),

heating from ice formation ($Q_{ice}$), sensible heating from the surface ($Q_{sen}$), surface evaporation ($Q_{evap}$), turbulent diffusion

of heat ($Q_{diff}$), and turbulent transport of water vapor ($Q_{H_2O}$) following

$$Q_{dia}(\theta_e, t) = Q'_{int}(\theta_e, t) + Q_{ice}(\theta_e, t) + Q_{sen}(\theta_e, t) + Q_{evap}(\theta_e, t) + Q_{diff}(\theta_e, t) + Q_{H_2O}(\theta_e, t) \qquad (4)$$

The terms $Q_{evap}$ and $Q_{H_2O}$ are expressed as heating rates by multiplying the underlying water fluxes by $L_v(T)/C_{pd}$. In order

to quantify the dominant processes contributing to temporal variation of $M_{\theta e}$, the terms in Eqs. 3 and 4 must be linked to

diagnostic variables available in the reanalysis or model products. Although there was no perfect match with any of the three

reanalysis products, MERRA-2 provides temperature tendencies for individual processes, which can be converted to heating

rates per Eq. 4 following

$$\frac{\partial Q_i(\theta_e, t)}{\partial \theta_e} = \frac{C_{pd}}{\Delta \theta_e} \sum_x \left(\frac{dT}{dt}\right)_{x,i} M_x \qquad (5)$$

where i refers a specific process ($Q'_{int}$, $Q_{ice}$, etc.), $\left(\frac{dT}{dt}\right)_x$ (K s$^{-1}$) is the temperature tendency of grid cell x, $M_x$ (kg) is the mass

of grid cell x, and $\Delta \theta_e$ is the width of the $\theta_e$ surface.

There are 5 heating terms provided in the MERRA-2 product, which we can approximately relate to terms in Eq. 4, as shown

in Table 1. The first three terms ($Q_{rad}$, $Q_{dyn}$, and $Q_{ana}$) can be summed to yield $Q'_{int}$, the forth ($Q_{trb}$) is equal to the sum of

$Q_{diff}$ and $Q_{sen}$, and the fifth ($Q_{mst}$) approximates the sum of $Q_{ice}$ and $Q_{evap}$. MERRA-2 does not provide terms corresponding

to $Q_{H_2O}$ or $Q_{evap}$ but $Q_{mst}$ represents heating due to moist processes, which includes $Q_{ice}$ plus water vapor

evaporation/condensation within the atmosphere. This water vapor evaporation/condensation should be approximately equal

to $Q_{evap}$ with small time lag when integrated over a $\theta_e$ surface because mixing is preferentially along the $\theta_e$ surface and water

vapor released into a $\theta_e$ surface by surface evaporation will tend to transport and precipitate from the same $\theta_e$ surface within a

short time period (Bailey et al., 2019). Thus, the MERRA-2 term for heating by moist processes should approximate $Q_{ice}$ +

$Q_{evap}$.

Figure 6a compares the temporal variation of $\dot{M}_{\theta_e}$ computed by integrating dry air mass (i.e., $\theta_e$-$M_{\theta e}$ look-up table) with $M_{\theta e}$

computed from the sum of the diabatic heating terms from MERRA-2 (via Eq. 3 to Eq. 5). The comparison focuses on the $\theta_e$

= 300 K surface, which does not intersect with the Equator or tropopause, so that the two mass flux terms ($m_T$, $m_E$) vanish.

These two methods have a high correlation at 0.71. We do not expect perfect agreement because $\dot{M}_{\theta_e}$ computed by the sum of

heating neglects turbulent water vapor transport, and only approximates $Q_{evap}$ as discussed above. This relatively good


agreement nevertheless demonstrates that the formulation based on MERRA-2 heating terms includes the dominant processes that drive temporal variations in the look-up table.

Figure 6b further breaks down the sum of the heating terms in Eq. 3 and 4 from MERRA-2 into individual components. Each term clearly displays variability on synoptic to seasonal scales. To quantify the contribution of different terms on the different time scales, we separate each term into a seasonal and synoptic component, where the seasonal component is derived by a two-harmonic fit with constant offset and the synoptic component is the residual. We estimate the fractional contribution of each heating term on seasonal and synoptic time scales separately in Table 2, using the method in Supplementary S2. On the seasonal

time scale, the variance is dominated by radiative heating and cooling of the atmosphere and the moist processes (including both ice formation and extra water vapor from surface evaporation) together, with prominent counteraction between each other. On the synoptic time scale, dynamic dissipation of energy dominates the variance.

We also carried out similar analyses on different $\theta_e$ surfaces (not shown). In all cases, we found that the combination of radiative heating and moist processes dominates the temporal variation of $M_{\theta e}$ on the seasonal time scale, while dynamic

dissipation of energy dominates on the synoptic time scale.

## 4 Applications of $M_{\theta e}$ as an atmospheric coordinate

To illustrate the potential application of $M_{\theta e}$ for interpreting sparse data, we focus on the seasonal cycle of $CO_2$ in the Northern Hemisphere as resolved by two series of global airborne campaigns, HIPPO and ATom. HIPPO consisted of five campaigns between 2009 and 2011 and ATom consisted of four campaigns between 2016 and 2018. Each campaign covered from ~ 150

m to ~ 14000 m and from nearly Pole to Pole, along both northbound and southbound transects. On HIPPO, both transects were over the Pacific Ocean, while on ATom, southbound transects were over the Pacific Ocean and northbound transects were over the Atlantic Ocean. The flight tracks are shown in Figure 7a. We aggregate data from each campaign into northbound and southbound transects within each hemisphere, but only use data from the Northern Hemisphere. We only consider tropospheric observations by excluding measurements from the stratosphere, which is defined by observed water vapor less

than 50 ppm and either $O_3$ greater than 150 ppb or detrended $N_2O$ to the reference year of 2009 less than 319 ppb. Water vapor and $O_3$ were measured by the NOAA UCATS (UAS Chromatograph for Atmospheric Trace Species) instrument and were interpolated to 10-sec resolution. $N_2O$ was measured by the Harvard QCLS (Quantum Cascade Laser System) instrument. Furthermore, we exclude all near-surface observations during take-offs, landings, and missed approaches, which usually show high $CO_2$ variability due to strong local influences. In-situ measurements of $CO_2$ were made by 3 different instruments on both

HIPPO and Atom. Of these, we use the $CO_2$ measurements made by the NCAR Airborne Oxygen Instrument (AO2) with a 2.5 seconds measurement interval (Stephens et al., submitted to AMTD, 2020), for consistency with planned future applications to APO (atmospheric potential oxygen) computed from AO2. The differences between instruments are small for our application (Santoni et al., 2014). The data used in this study are averaged to 10-sec resolution and we show the detrended $CO_2$ values along each airborne campaign transect for the Northern Hemisphere in Figure 7b, Since we focus on the seasonal

cycle of $CO_2$, all airborne observations are detrended by subtracting an interannual trend fitted to $CO_2$ measured at the Mauna Loa Observatory (MLO) by the Scripps $CO_2$ Program. This trend is computed by a stiff cubic spline function plus 4-harmonic terms with linear gain to the MLO record. $M_{\theta e}$ is computed from ERA-Interim in this section.

## 4.1 Mapping Northern Hemisphere $CO_2$

A conventional method to display seasonal variations in $CO_2$ from airborne data is to plot time series of the data at a given
location or latitude and different pressure levels (Graven et al., 2013; Sweeney et al., 2015). In Figure 8, we compare this method using HIPPO and ATom airborne data, binning and averaging the data from each airborne campaign transect by pressure and latitude bins, with our new method, binning the data by pressure and $M_{\theta e}$. For each latitude bin, we choose a corresponding $M_{\theta e}$ bin which has approximately the same meridional coverage in the lower troposphere. We remind the reader that $M_{\theta e}$ decreases poleward, while also generally increasing with altitude (Figures 2 and 3).

As shown in Figure 8, the transect average of detrended $CO_2$ (shown as points) from both binning methods resolve well-defined seasonal cycles (based on 2-harmonic fit) in all bins, with higher amplitudes near the surface (low pressure) and at high latitude/ low $M_{\theta e}$. However, binning by $M_{\theta e}$ leads to much smaller variations of the mean seasonal cycle (shown as solid curves) with pressure, as expected, because moist isentropes are preferential surfaces of mixing. Also, within individual pressure bins, the short-term variability relative to the mean cycles based on the distribution of all detrended observations (not
shown as points but denoted as 1 σ values in Figure 8) is smaller when binning by $M_{\theta e}$ (F-test, $p < 0.01$), except in the lower troposphere of the highest $M_{\theta e}$ bin (90-110 $10^{16}$ kg). The smaller short-term variability is expected because $M_{\theta e}$ tracks the synoptic variability of the atmosphere. When binning by latitude, the smallest short-term variability is found at the lowest bin (surface-800 mbar) and the largest short-term variability is found in the highest bin (500 mbar-tropopause), except the highest latitude bin (45° N-55° N). When binning by $M_{\theta e}$, in contrast, the short-term variability in the middle pressure bin is always
smaller than the higher and lower pressure bins (F-test, $p < 0.01$), except for the 50 to 70 $M_{\theta e}$ bin, where the difference between the lowest and middle pressure bins is not significant (based on 1 σ levels). The lower variability in the mid troposphere may reflect the suppression of variability from synoptic disturbances, leaving a clearer signal of the influence of surface fluxes of $CO_2$ and stratosphere-troposphere exchanges. We compare the variance of detrended airborne observations within each $M_{\theta e}$-pressure bin with its fitted value. The fitted seasonal cycle of each bin explains 63.2% to 90.5% of the variability for different
bins, with higher fractions in the middle troposphere.

Figure 8 also shows the $CO_2$ seasonal cycle at MLO, which falls within a single $M_{\theta e}$-pressure bin (90-110 $10^{16}$ kg, 500-800 mbar) at all seasons. Although the airborne data in this bin span a wide range of latitudes (~ 10° N-75° N), the seasonal cycle averaged over this bin is very similar to the cycle at MLO (airborne cycle leads by ~10 days with 1.0% lower amplitude).

It is also of interest to examine how $CO_2$ data from surface stations fit into the framework based on $M_{\theta e}$. Figure 9 compares
the $CO_2$ seasonal cycle of five NOAA surface stations (Dlugokencky et al., 2019) with the cycle from the airborne observations





binned into selected $M_{\theta e}$ bins. These surface stations are chosen to be representative of different $M_{\theta e}$ ranges. For the comparison, we chose $M_{\theta e}$ bins that span the seasonal maximum and minimum $M_{\theta e}$ value of the station. These bins are narrower than the bins used in Figure 8, in order to sharply focus on the latitude of the station. To maximize sampling coverage, we bin the airborne data only by $M_{\theta e}$ without pressure sub-bins. For mid- and high latitude surface stations (right three panels), the

seasonal amplitude of station $CO_2$ and corresponding airborne $CO_2$ are close (within 4−5%), while airborne cycles lag by 2−3 weeks. The lag presumably represents the slow mixing from the mid-latitude surface to the high latitude mid-troposphere (Jacob, 1999). In contrast, for low latitude stations (left two panels) which generally sample trade winds, the seasonal cycles differ significantly, indicating that the air sampled at these stations is not rapidly mixed along surfaces of constant $M_{\theta e}$ or $\theta_e$ with air aloft. As mentioned above (Section 3.1), surfaces of high $M_{\theta e}$ within the Hadley circulation have two branches, one

near the surface and one aloft. A timescale of several months for transport from the lower to the upper branch can be estimated from the known overturning flows based on air mass flux streamfunctions (Dima and Wallace, 2003). This delay, plus strong mixing and diabatic effects (Miyazaki et al., 2008), ensures that the lower and upper branches are not well connected on seasonal time scales. Our results nevertheless demonstrate that the $M_{\theta e}$ framework combining airborne and surface data could help understand details of atmospheric transport both along and across $\theta_e$ surfaces.

**4.2 Computing the hemispheric mass-weighted average $CO_2$ mole fraction**

We next illustrate the use of $M_{\theta e}$ for computing the mass-weighted average of a long-lived chemical tracer by performing this exercise for $CO_2$ in the Northern Hemisphere. We calculate the Northern Hemisphere tropospheric mass-weighted average $CO_2$ from each airborne transect using a method that assumes that $CO_2$ is uniformly mixed on $\theta_e$ surfaces throughout the hemisphere. HIPPO-1 Northbound is excluded here due to the lack of data north of 40° N. We use the $\theta_e$-$M_{\theta e}$ lookup table of

the corresponding date to assign a value of $M_{\theta e}$ to each observation based on its $\theta_e$. The observations for each transect are then sorted by $M_{\theta e}$. The hemispheric average $CO_2$ is calculated by trapezoidal integration of $CO_2$ as a function of $M_{\theta e}$ and divided by the total dry air mass as computed from the corresponding range of $M_{\theta e}$.

To illustrate the $M_{\theta e}$ integration method, we choose HIPPO-1 Southbound and show $CO_2$ measurements and $\Delta CO_2$ inventory (Pg) as a function of $M_{\theta e}$ in Figure 10. The Northern Hemisphere tropospheric average detrended $CO_2$ is computed by

integrating the area under the curve (subtracting negative contributions) and dividing by the maximum value of $M_{\theta e}$ within the hemisphere (here $195.13 \times 10^{16}$ kg). This yields a mass-weighted average detrended $CO_2$ of 1.13 ppm for the full troposphere of the Northern Hemisphere. The trapezoidal integration has a high accuracy because the data are dense over $M_{\theta e}$. The $\Delta CO_2$ inventory is dominated by the domain $M_{\theta e} < 120$, with less than 4.1% contributed by the additional ~38.8% of the air mass outside this domain (Fig. 10b).

We compute a Northern Hemisphere mass-weighted average detrended $CO_2$ for each airborne campaign transect and fit the time series to a 2-harmonic fit to estimate the seasonal cycle (Figure 11). We find that the cycle has a seasonal amplitude of





7.9 ppm and a downward zero-crossing at Julian day 179, where the latter is defined as the date when the detrended seasonal cycle changes from positive to negative.

To address the error in fitted amplitude and zero crossing, we consider two main sources: (1) irreproducibility in the $CO_2$ measurements and (2) limited coverage in space and time. For the first contribution, we compute the difference between mass-weighted average $CO_2$ from AO2 and mean mass-weighted average $CO_2$ from Harvard QCLS, Harvard OMS, and NOAA Picarro for each airborne campaign transect, while masking values that are missing in any of these datasets. We compute the standard deviation of these differences (± 0.15 ppm) for mass-weighted average $CO_2$ of each airborne campaign transect as the 1σ level of uncertainty. We further compute the uncertainties for the seasonal amplitude of ± 0.11 ppm and for the

downward zero-crossing of ± 0.83 days, which are calculated from 1000 iterations of the 2-harmonic fit, allowing for random Gaussian uncertainty (σ = ± 0.11 ppm) for each transect.

    For the contribution to the error in the amplitude and phase from limited space/time coverage, we use simulated $CO_2$ data from the Jena $CO_2$ Inversion Run ID: s04oc v4.3 (Rödenbeck et al., 2003, 2018). This model includes full atmospheric fields from 2009 to 2018, which we detrend using the cubic spline fit to the observed MLO trend. From these detrended fields, we compute

the climatological cycle of the Northern Hemisphere average by integrating over all tropospheric grid cells (cutoff at PVU = 2) to produce a daily time series of the hemispheric mean, which we take as the model "truth". We fit a 2-harmonic function to this "true" time series to compute a "true" climatological cycle over the 2009-2018 period (Table 3), which is our target for validation. We then subsample the Jena $CO_2$ Inversion along the HIPPO and ATom flight tracks and process the data similarly to the observations, using the $M_{\theta e}$ integration method and a 2-harmonic fit. The comparison shows that the $M_{\theta e}$ integration

method yields an amplitude which is 1% too large and yields a downward zero-crossing date which is 6 days too late. We view these offsets as systematic biases, which we correct from the observed amplitude and phase reported above. The uncertainties in these biases are hard to quantify, but we take ±100 % as a conservative estimate. We thus allow an additional random error of ± 0.08 ppm in amplitude and ± 6.0 days in downward zero crossing for uncertainty in the bias. Combining the random and systematic error contributions leads to a corrected Northern Hemisphere tropospheric average $CO_2$ seasonal cycle amplitude

of 7.8 ± 0.14 ppm and downward zero-crossing of 173 ± 6.1 days. This corrected cycle is an estimate of the climatological average from 2008-2019.

    The error due to limited space/time coverage can be divided into three components: limited seasonal coverage (17 transects over the climatological year), limited interannual coverage (sampling particular years instead of all years), and limited spatial coverage (under-sampling the full hemisphere). We quantify the combined biases due to both limited seasonal and interannual

coverage by comparing the two-harmonic fit of the full "true" daily time series of the hemispheric mean to a two-harmonic fit of that data subsampled on the actual mean sampling dates of the 17 flight tracks. We isolate the bias associated with limited seasonal coverage by repeating this calculation, replacing the "true" daily time series with the daily climatological cycle. The bias associated with limited spatial coverage is quantified as the residual. Combining these results, we estimate that the limited





seasonal, interannual, and spatial coverage, account for biases in the downward zero-crossing of 1.1, 1.4, and 3.5 days
respectively, all in the same direction (too late). The seasonal amplitude bias due to individual components are all small (<
0.5%).

It is of some interest to compare our estimate of the Northern Hemisphere average cycle with the cycle at Mauna Loa, which
is also broadly representative of the hemisphere. Our comparison in Figure 11 shows small but significant differences in both
amplitude and phase, with the MLO amplitude being ~ 11.5% smaller than the hemispheric average and lagging in phase by
~ 1 month. There are also differences in the shape of the cycle, with the MLO cycle rising more slowly from October to
February, but more quickly from February to May. These features at least partly reflects variations in the transport of air masses
to the station (Harris et al., 1992; Harris and Kahl, 1990).

In Figure 12, we compare the $M_{\theta e}$ integration method with an alternate latitude-pressure weighted average method, with no
correction for synoptic variability. For this method, we bin flight track subsampled Jena $CO_2$ Inversion data into sin(latitude)-
pressure bins with 0.01 and 25 mbar as intervals respectively, while all bins without data are filtered. We further compute a
weighted average $CO_2$ for each airborne campaign transect. The root-mean-square errors (RMSE) to the true average of the
$M_{\theta e}$ integration method are $\pm 0.32$ and $\pm 0.27$ ppm for HIPPO and ATom campaigns, respectively, which are smaller than the
RMSE of the simple latitude-pressure weighted average method at $\pm 0.82$ and $\pm 0.53$ ppm.

We also evaluate the biases in the hemispheric average season cycles computed with the simple latitude-pressure weighted
average method. As summarized in Table 3, the latitude-pressure weighted average method yields a larger error in seasonal
amplitude ($M_{\theta e}$ method 1.0 % too large, latitude-pressure method 20.8% too large), while both methods show similar phasing
error (6 to 7 days late). The larger error associated with the latitude-pressure weighted average method is consistent with strong
influence of synoptic variability. This synoptic variability could potentially be corrected using model simulations of the 3-
dimensional $CO_2$ fields (Bent, 2014). The $M_{\theta e}$ integration method appears advantageous because it accounts for synoptic
variability, and easily yields a hemispheric average by directly integrating over $M_{\theta e}$.

The relative success of the $M_{\theta e}$ integration method in yielding accurate hemispheric averages using HIPPO and ATom data is
attributable partly to the extensive data coverage. To explore the coverage requirement for reliably resolving hemispheric
averages, we also test the integration method when applied to simulated data with lower coverage. We start with the same
coverage as for ATom and HIPPO but select only subsets of the points in four groups: poleward of 30° N, Equator to 30° N,
surface to 600 mbar, and 600 mbar to tropopause. We also examine whether we can only utilize observation along the Pacific
transect by excluding measurements along the Atlantic transects (ATom northbound). We further explore the impact of reduced
sampling density by subsampling the Jena $CO_2$ Inversion based on the spatial coverage of the Medusa sampler, which is an
airborne flask sampler that collected 32 cryogenically dried air samples per flight during HIPPO and ATom (Stephens et al.,
submitted to AMTD, 2020). We further randomly retain 10%, 5%, and 1% of the full flight track subsampled data, repeating
each ratio with 1000 iterations. We compute the detrended average $CO_2$ from these nine simulations by the $M_{\theta e}$ integration





method and then compute the RMSE relative to the detrended true hemispheric average, together with the seasonal magnitude and day of year of the downward zero-crossing, as summarized in Table 3. HIPPO-1 Northbound is excluded in all these simulations. The number of data points of each simulation and number of observations of the original HIPPO and ATom data sets are summarized in Table S1. These results show that limiting sampling to either equatorward or poleward of 30° N yields

significant error (24.3% smaller and 24.9% larger seasonal amplitude, respectively). Additionally, there is a ~ 25 days lag in phase if sampling is limited to equatorward of 30° N. However, restricting sampling to be exclusively above or below 600 mbar, or only along the Pacific transect does not lead to significant errors. Randomly reducing the sampling by 10- to 100- fold or only keeping Medusa spatial coverage also have minimal impact. This suggests that, to compute the average $CO_2$ of a given region, it may be sufficient to have low sampling density provided that the measurements adequately cover the full range

in $\theta_e$ (or $M_{\theta e}$).

## 5 Discussion and summary

We have presented a transformed isentropic coordinate, $M_{\theta e}$, which is the total dry air mass under a given $\theta_e$ surface in the troposphere of the hemisphere. $M_{\theta e}$ can be computed from meteorological fields by integrating dry air mass over $\theta_e$ surfaces, and different reanalysis products show a high consistency. The $\theta_e$-$M_{\theta e}$ relationship varies seasonally due to seasonal

heating/cooling of the atmosphere via radiative heating and moist processes. The seasonality in the relationship is greater at low $\theta_e$ compared to high $\theta_e$, and is greater in the Northern than the Southern Hemisphere. The $\theta_e$-$M_{\theta e}$ relationship also shows synoptic-scale variability, which is mainly driven by dynamic dissipation of energy. $M_{\theta e}$ surfaces show much less seasonal displacement with latitude and altitude than surfaces of constant $\theta_e$, while being parallel and exhibiting essentially identical synoptic scale variability. As a coordinate for mapping tracer distributions, $M_{\theta e}$ shares with $\theta_e$ the advantages of following

displacements due to synoptic disturbances and aligning with surfaces. $M_{\theta e}$ has the additional advantage of being approximately fixed in space seasonally, which allows mapping to be done on seasonal time scales, and having units of mass, which provides a close connection with atmospheric inventories.

As a coordinate, $M_{\theta e}$ is probably better viewed as an alternative to latitude, due to its nearly fixed relationship with latitude over season, rather than as an alternative to altitude (or pressure), as typically done for potential temperature (Miyazaki et al.,

2008; Miyazaki and Iwasaki, 2005; Parazoo et al., 2011; Tung, 1982; Yang et al., 2016). Even though the contours of constant $M_{\theta e}$ extend over a wide range of latitudes (from low latitudes at the Earth surface to high latitudes aloft), a close association with latitude is provided by the point of contact with Earth's surface. Also, $M_{\theta e}$ is nearly always monotonic with latitude (increasing equatorward) while it is not necessarily monotonic with altitude in the lower troposphere (Figure 2).

As a first application, we have illustrated using $M_{\theta e}$ to map the seasonal variation of $CO_2$ in the Northern Hemisphere, using

data from the HIPPO and ATom airborne campaigns. This application shows that $M_{\theta e}$ has several advantages as a coordinate compared to using latitude: (1) variations in $CO_2$ with pressure are smaller at fixed $M_{\theta e}$ than at fixed latitude, and (2) the scatter about the mean $CO_2$ seasonal cycle is smaller when sorting data into pressure/$M_{\theta e}$ bins than into pressure/latitude bins. We



have also shown that, at middle and high latitudes, the $CO_2$ seasonal cycles that are resolved in the airborne data (binned by $M_{\theta e}$ but not pressure) are very similar to the cycles observed at surface stations at the appropriate latitude, with a phase lag of
~ 2 to 3 weeks. At lower latitudes, $CO_2$ cycles in the airborne data (binned similarly by $M_{\theta e}$) are less consistent with surface data, as expected due to slow transport and diabatic processes within the Hadley Circulation. For characterizing the patterns of variability in airborne $CO_2$ data, we expect the advantages of $M_{\theta e}$ over latitude will be greatest for sparse datasets, allowing data to be binned more coarsely with pressure or elevation while still resolving features of large-scale variability, such as seasonal cycles or gradients with latitude.

As a second application, we use $M_{\theta e}$ to compute the Northern Hemisphere tropospheric average $CO_2$ from the HIPPO and ATom airborne campaigns by integrating $CO_2$ over $M_{\theta e}$ surfaces. With a small correction for systematic biases induced by limited hemispheric coverage of the HIPPO and ATom flight tracks, we report a seasonal amplitude of $7.8 \pm 0.14$ ppm and a downward zero-crossing at Julian day $173 \pm 6.1$. This hemispheric average cycle may prove valuable as a target for validation of models of surface $CO_2$ exchange.

Our analysis also clarifies that computing hemispheric averages with the $M_{\theta e}$ integration method depends on adequate spatial coverage. The coverage provided by the HIPPO and ATom campaigns appears more than adequate for computing the average seasonal cycle of $CO_2$ in the Northern Hemisphere, and the errors for this application remain small if the coverage is limited to either above or below 600 mbar, or reduced to retain only 1% of the measurements. Most critical is maintaining coverage in latitude, or $M_{\theta e}$ surfaces. The $M_{\theta e}$ integration method of computing hemispheric averages assumes that the tracer is uniformly
distributed and instantly mixed on $\theta_e$ ($M_{\theta e}$) surfaces. We have shown that systematic gradients in $CO_2$ are resolved with pressure at fixed $M_{\theta e}$, which reflects the finite rates of dispersion on $\theta_e$ surfaces. Further improvements to the integration method seem possible by integrating separately over different pressure levels, taking account of the different mass fraction in different pressure bins (e.g. Figure 4). The need is especially relevant for high $M_{\theta e}$ bins which are less completely mixed, and which tend to intersect the Equator or have separate surface branches. For these $M_{\theta e}$ bins, it would be more appropriate to integrate
over $M_\theta$ in the upper and lower atmosphere separately. This complication is of minor importance for computing the mass-weighted average $CO_2$ cycle, because the cycle of $CO_2$ is small in these air masses.

The definition of $M_{\theta e}$ requires horizontal and vertical boundaries for the integration of dry air mass. We use the dynamic tropopause (based on PVU) and the Equator as boundaries, which is appropriate for integrating tropospheric inventories in a hemisphere. Other boundaries may be more appropriate for other applications. For example, $M_{\theta e}$ could be computed from the
lowest $\theta_e$ surface in the Southern Hemisphere with a latitude cutoff at 30° S, to apply to airborne observations only over the Southern Ocean. On the other hand, the boundary choice only influences $M_{\theta e}$ surfaces that actually intercept the boundaries, making the choice less important at high latitude in the lower troposphere (lowest $M_{\theta e}$ surfaces). Some tropospheric applications may also benefit by integrating over dry potential temperature ($\theta$) rather than $\theta_e$.



Based on our promising results for $CO_2$, we expect that $M_{\theta e}$ may be usefully applied as a coordinate for mapping and computing
inventories of many tracers, such as $O_2/N_2$, $N_2O$, $CH_4$, and the isotopes of $CO_2$, whose residence time is long compared to the
time scale for mixing along isentropes. $M_{\theta e}$ may also prove useful in the design phase of airborne campaigns to ensure strategic
coverage. Our results show that, to study the seasonal cycle of a tracer on a hemispheric scale, it is critical to have well-
distributed sampling in $M_{\theta e}$.

## 6 Code availability

We provide R code to generate $\theta_e$-$M_{\theta e}$ look-up tables from ERA-Interim meteorological fields at
https://github.com/yumingjin0521/Mtheta.

## 7 Data availability

All HIPPO 10-sec merge data are available from: https://doi.org/10.3334/CDIAC/HIPPO_010 (Wofsy et al., 2017b). Besides,
all HIPPO Medusa merge data are available from: http://dx.doi.org/10.3334/CDIAC/hippo_014 (Wofsy et al., 2017a). All
ATom 10-sec and Medusa merges data are available from: https://doi.org/10.3334/ORNLDAAC/1581 (Wofsy et al., 2018).

$CO_2$ data from Mauna Loa Observatory are available from the Scripps $CO_2$ Program at: https://scrippsco2.ucsd.edu. Other
surface station $CO_2$ data, including Trinidad Head, Cold Bay, Barrow, Cape Kumukahi, Sand Island are provided by
NOAA/ESRL GMD flask sampling network (http://www.cmdl.noaa.gov/ccgg/trends) and downloaded from Observation
Package (ObsPack) at http://dx.doi.org/10.25925/20190812 (Dlugokencky et al., 2019).

The Jena $CO_2$ Inversion are available at the project website: http://www.bgc-jena.mpg.de/CarboScope/s/main.html. Run ID:
s04oc v4.3 was used in this study.

ERA-Interim is available at: https://www.ecmwf.int/en/forecasts/datasets/reanalysis-datasets/era-interim. NCEP2 is available
at: https://psl.noaa.gov/data/gridded/data.ncep.reanalysis2.html. MERRA-2 is available at the NASA Goddard Earth Sciences
(GES) Data and Information Services Center (DISC) at: https://disc.gsfc.nasa.gov/datasets?keywords=%22MERRA-
2%22&page=1&source=Models%2FAnalyses%20MERRA-2.

$\theta_e$-$M_{\theta e}$ look-up tables with daily resolution and 1 K intervals in $\theta_e$ from 1980 to 2018 computed from ERA-Interim are available
at https://github.com/yumingjin0521/Mtheta.

## 8 Appendix A: Temporal variation of $M_{\theta e}$

Following Walin's derivation for cross-isothermal volume flow in the ocean (Walin, 1982), we show how $\dot{M}_{\theta_e} = \frac{\partial}{\partial t} M_{\theta_e}(\theta_e, t)$
can be related to energy and mass fluxes. We start by deriving the relationship for $M_\theta$ (based on potential temperature $\theta$) but
later generalize to apply to $M_{\theta e}$.





All definitions are summarized in Table A1, and Figure A1 is the schematic diagram of mass and energy flux.

All mass and heat fluxes are counted positive as into region R(θ, t). The heat fluxes through tropopause, Equator and surface of region R(θ, t) can be divided into an advective and a turbulent component, D(θ, t). Integrating over the tropopause and
equatorial boundary, we have:

$$Q_T(\theta, t) = C_{pd} \int_{-\infty}^{\theta} \frac{\partial F_T(\theta', t)}{\partial \theta'} \theta' d\theta' + \int_{-\infty}^{\theta} \frac{\partial D_T(\theta', t)}{\partial \theta'} d\theta' \tag{A1}$$

$$Q_E(\theta, t) = C_{pd} \int_{-\infty}^{\theta} \frac{\partial F_E(\theta', t)}{\partial \theta'} \theta' d\theta' + \int_{-\infty}^{\theta} \frac{\partial D_E(\theta', t)}{\partial \theta'} d\theta' \tag{A2}$$

$$Q_I(\theta, t) = C_{pd} \cdot F_I(\theta, t) \cdot \theta + D_I(\theta, t) \tag{A3}$$

where $C_{pd}$ is the heat capacity of dry air in units of J kg$^{-1}$ K$^{-1}$.

Based on the continuity of mass and energy for region R(θ, t), we obtain

$$\frac{\partial}{\partial t} M_\theta(\theta, t) = F_T(\theta, t) + F_E(\theta, t) + F_I(\theta, t)$$

$$= \int_{-\infty}^{\theta} \frac{\partial F_T(\theta', t)}{\partial \theta'} d\theta' + \int_{-\infty}^{\theta} \frac{\partial F_E(\theta', t)}{\partial \theta'} d\theta' + F_I(\theta, t) \tag{A4}$$

$$C_{pd} \frac{\partial}{\partial t} \int_{-\infty}^{\theta} \frac{\partial M_\theta(\theta', t)}{\partial \theta'} \theta' d\theta' = Q_T(\theta, t) + Q_E(\theta, t) + Q_I(\theta, t) + \int_{-\infty}^{\theta} \frac{\partial Q_s(\theta', t)}{\partial \theta'} d\theta' + \int_{-\infty}^{\theta} \frac{\partial Q_{int}(\theta', t)}{\partial \theta'} d\theta' \tag{A5}$$

Substituting Eq. A1 to Eq. A3 into Eq. A5 and differentiating with respect to θ yields

$$C_{pd}\theta \frac{\partial}{\partial t} \frac{\partial M_\theta(\theta, t)}{\partial \theta} = C_{pd}\theta \left( \frac{\partial F_T(\theta, t)}{\partial \theta} + \frac{\partial F_E(\theta, t)}{\partial \theta} + \frac{\partial F_I(\theta, t)}{\partial \theta} \right) + C_{pd}F_I(\theta, t) +$$


$$\frac{\partial Q_{diff}(\theta, t)}{\partial \theta} + \frac{\partial Q_s(\theta, t)}{\partial \theta} + \frac{\partial Q_{int}(\theta, t)}{\partial \theta} \tag{A6}$$

where,

$$Q_{diff}(\theta, t) = \int_{-\infty}^{\theta} \frac{\partial D_T(\theta', t)}{\partial \theta'} d\theta' + \int_{-\infty}^{\theta} \frac{\partial D_E(\theta', t)}{\partial \theta'} d\theta' + D_I(\theta, t) \tag{A7}$$

Differentiating Eq. A4 with respect to θ, and multiplying $C_{pd} \cdot \theta$ yields

$$C_{pd}\theta \frac{\partial}{\partial t} \frac{\partial M_\theta(\theta, t)}{\partial \theta} = C_{pd}\theta \left( \frac{\partial F_T(\theta, t)}{\partial \theta} + \frac{\partial F_E(\theta, t)}{\partial \theta} + \frac{\partial F_I(\theta, t)}{\partial \theta} \right) \tag{A8}$$

Subtracting Eq. A8 from Eq. A6, we obtain





$$C_{pd}F_I(\theta, t) = -\frac{\partial Q_{diff}(\theta, t)}{\partial \theta} - \frac{\partial Q_s(\theta, t)}{\partial \theta} - \frac{\partial Q_{int}(\theta, t)}{\partial \theta} \tag{A9}$$

Eq. A9 divided by $C_{pd}$ plus Eq. A4 yields

$$\frac{\partial}{\partial t}M_\theta(\theta, t) = -\frac{1}{C_{pd}}\left(\frac{\partial Q_{diff}(\theta, t)}{\partial \theta} + \frac{\partial Q_s(\theta, t)}{\partial \theta} + \frac{\partial Q_{int}(\theta, t)}{\partial \theta}\right) + \int_{-\infty}^{\theta}\frac{\partial F_T(\theta', t)}{\partial \theta'}d\theta' + \int_{-\infty}^{\theta}\frac{\partial F_E(\theta', t)}{\partial \theta'}d\theta' \tag{A10}$$

Eq. A10 illustrates the temporal variation of $M_\theta$, where $Q_{int}$ includes radiative heating (i.e. sum of shortwave and longwave
heating), dynamic dissipation of heat, and latent heat releasing due to evaporation and condensation.

To modify Eq. A10 to apply to $M_{\theta e}$ rather than $M_\theta$, it is necessary to replace all $\theta$ with $\theta_e$, and additionally account for the
following:

1. Condensation and evaporation is conserved on the $\theta_e$ surfaces, but the gaining and losing of water vapor through surface
evaporation and water vapor transport contributes to $\theta_e$. This contribution can be computed as the product of latent heat of
evaporation and the extra water vapor content. Thus, the surface contribution ($Q_S$) needs to include both sensible heating of
the atmosphere ($Q_{sen}$) and the water vapor flux from the surface into the atmosphere ($Q_{evap}$). Similarly, the diffusion term
within the atmosphere ($Q_{diff}$) needs to include both heat and water vapor ($Q_{H_2O}$).

2. Internal heating ($Q_{int}$) needs to exclude latent heat releasing due to evaporation and condensation of liquid water, which
cancel in $\theta_e$, but it still needs to include heating from ice formation, which does not cancel in $\theta_e$. We subtract this ice component
from the rest of the internal heating, yielding two terms $Q'_{int}$ and $Q_{ice}$, with $Q_{int} = Q'_{int} + Q_{ice}$.

Therefore, we can write the temporal variation of $M_{\theta e}$ as

$$\frac{\partial}{\partial t}M_{\theta e}(\theta_e, t) = \int_{-\infty}^{\theta_e}\frac{\partial F_T(\theta'_e, t)}{\partial \theta'_e}d\theta'_e + \int_{-\infty}^{\theta_e}\frac{\partial F_E(\theta'_e, t)}{\partial \theta'_e}d\theta'_e -$$
$$\frac{1}{C_{pd}}\left(\frac{\partial Q_{diff}(\theta_e, t)}{\partial \theta_e} + \frac{\partial Q_{sen}(\theta_e, t)}{\partial \theta_e} + \frac{\partial Q_{evap}(\theta_e, t)}{\partial \theta_e} + \frac{\partial Q'_{int}(\theta_e, t)}{\partial \theta_e} + \frac{\partial Q_{ice}(\theta_e, t)}{\partial \theta_e} + \frac{\partial Q_{H_2O}(\theta_e, t)}{\partial \theta_e}\right) \tag{A11}$$

## 9 Authors contributions

YJ carried out the data analysis and derivations. Initial drafts were prepared by YJ and RFK, with additional contributions
from all co-authors.

## 10 Competing interests

The authors declare that they have no conflict of interest.





## 11 Acknowledgements

The original $M_{\theta e}$ concept arose out of discussions during the ORCAS field campaign that included Ralph Keeling, Colm
Sweeney, Eric Kort, Matthew Long, and Martin Hoecker-Martinez. We would like to acknowledge the efforts of the full
HIPPO and ATom science teams and the pilots and crew of the NCAR/NSF GV and NASA DC-8, the NCAR and NASA
project managers, field support staff, and logistics experts. In this work, we have used the HIPPO and ATom 10-sec merge
files, supported by the National Center for Atmospheric Research (NCAR). NCAR is sponsored by the National Science
Foundation under Cooperative Agreement No. 1852977. The HIPPO program was supported by NSF grants ATM-0628575,
ATM-0628519 and ATM-0628388 to Harvard University, University of California San Diego, NCAR, and the University of
Colorado/CIRES. The ATom program was supported by the NASA grant NNX15AJ23G. Medusa and AO2 measurements on
ATom were supported NSF grants AGS-1547797 and AGS-1623748 to University of California San Diego and NCAR. YJ
and EJM are also supported under AGS-1623748. We thank the Harvard QCLS, Harvard OMS, NOAA UCATS and NOAA
Picarro teams for sharing measurements. We thank NOAA ESRL GML for providing surface station $CO_2$ data measured at
Trinidad Head, Cold Bay, Barrow, Cape Kumukahi, and Sand Island. We thank Christian Rödenbeck for sharing Jena $CO_2$
Inversion run.

Any opinions, findings, and conclusions or recommendations expressed in this material are those of the authors and do not
necessarily reflect the views of the National Science Foundation.





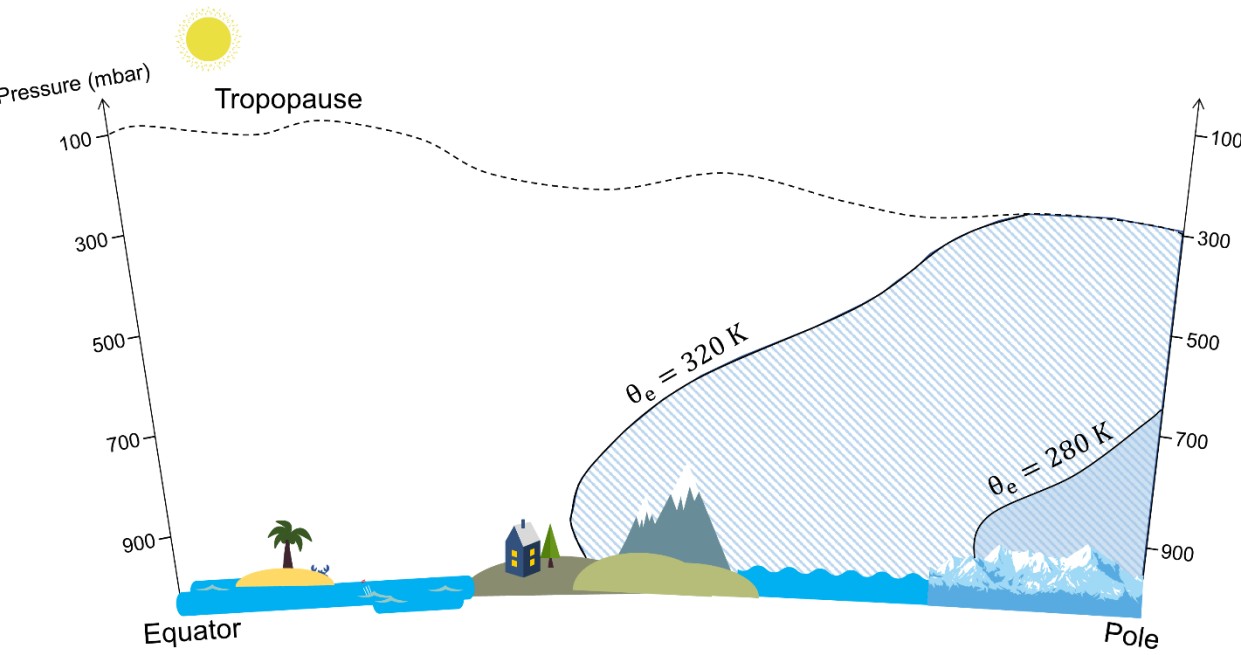


**Figure 1: Schematic of the conceptual basis to calculate $M_{\theta e}$. $M_{\theta e}$ of a given $\theta_e$ surface is computed by summing all dry air mass with a low equivalent potential temperature in the troposphere of the hemisphere. This calculation yields a unique $\theta_e$-$M_{\theta e}$ relation at a given time point.**





Figure 2: Snapshot of the distribution of (a) zonal average $\theta_e$ surfaces on 1 January 2009 (solid lines) and 1 July 2009 (dashed lines), (b) zonal average $M_\theta$ surfaces on 1 January 2009 (solid lines) and 1 July 2009 (dashed lines). The zonal average tropopause is also shown here for 1 January 2009 (solid black line) and 1 July 2009 (dashed black line). $\theta_e$, $M_{\theta e}$ and tropopause are computed from ERA-Interim.





**Figure 3: Time series of meridional displacement of selected zonal average θ_e (K) surfaces over a year at (a) 500 mbar, (b) 700 mbar and (c) 925 mbar. Meridional displacement of selected zonal average M_{θe} ($10^{16}$ kg) surfaces over a year at (d) 500 mbar, (e) 700 mbar and (f) 925 mbar. The value of each surface is labelled. θ_e and M_{θe} are computed from ERA-Interim. Results shown are for year 2009.**



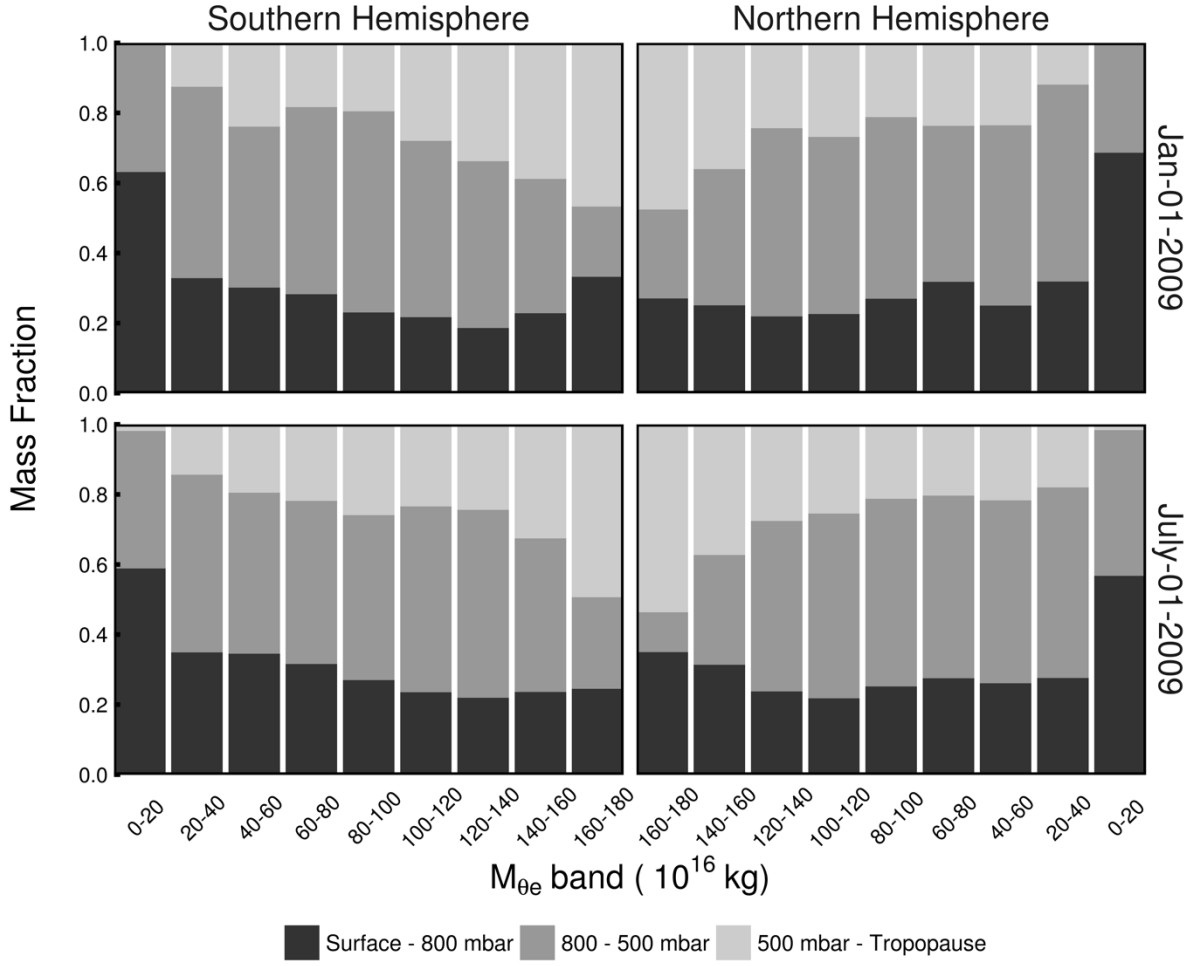

**Figure 4: Snapshots (1 January 2009 and 1 July 2009) of the mass distribution of different $M_{\theta e}$ bins from three pressure bins (surface to 800 mbar, 800 mbar to 500 mbar, and 500 mbar to tropopause). $M_{\theta e}$ is computed from ERA-Interim. Low $M_{\theta e}$ bins are seen to have larger contributions from the air near the surface, and high $M_{\theta e}$ bins have larger contributions from air aloft. Comparing the top and the bottom panels shows that the seasonal differences in pressure contributions are small except for the highest $M_{\theta e}$ bins (160-180) and the lowest $M_{\theta e}$ bin in the northern hemisphere (0-20).**





**Figure 5: Variability of $M_{\theta e}$ of given $\theta_e$ surfaces (i.e., $\theta_e$-$M_{\theta e}$ look-up table) over a year with daily resolution in the Northern and Southern Hemisphere. Data from ERA-Interim is shown as a solid line, MERRA-2 is shown as a dashed line and NCEP2 is shown as a dotted line. Results shown are for year 2009.**



**Figure 6: (a) Temporal variation of $M_{\theta e}$ in the Northern Hemisphere at $\theta_e = 300$ K computed by integrating air mass (blue line) and estimated from the sum of five heating terms (Table 1) in MERRA-2 (black line). (b) The heating variables decomposed into five contributions as indicated (see Table 1). Results shown are for year 2009.**





**Figure 7: (a) HIPPO and ATom horizontal flight tracks coloured by campaigns. (b) Latitude and pressure cross-section of detrended**
**$CO_2$ of each airborne campaign transect. $CO_2$ is detrended by subtracting MLO stiff cubic spline trend, which is computed by a stiff cubic spline function plus 4-harmonic functions with linear gain to MLO record.**



**Figure 8:** Seasonal cycles of airborne Northern Hemisphere $CO_2$ data sorted by (a) $M_{\theta e}$-pressure bins and (b) latitude-pressure bins. $M_{\theta e}$ bins ($10^{16}$ kg) and latitude bins are shown on the top of each panel. Pressure bins are coloured. The latitude bounds are chosen to approximate the meridional coverage of each corresponding $M_{\theta e}$ bin in the lower troposphere. The seasonal cycle at MLO from 2009 and 2018 is shown on the 90–110 $M_{\theta e}$ bin panel, which spans the $M_{\theta e}$ of the station. Airborne observations are first grouped into $M_{\theta e}$-pressure or latitude-pressure bins, and then averaged for each airborne campaign transect, shown as points. We filter out the points averaged from less than 20 10-sec observations. The seasonal cycle of airborne data and MLO (2009-2018) are computed by a 2-harmonic fit to the detrended time series. The $1\sigma$ variability about the seasonal cycle fits for each $M_{\theta e}$-pressure or latitude-pressure bin are labelled on top of each panel. These $1\sigma$ values are based on the distribution of all binned observations (not shown), rather than the distribution of average $CO_2$ of each bin and airborne campaign transect (shown).

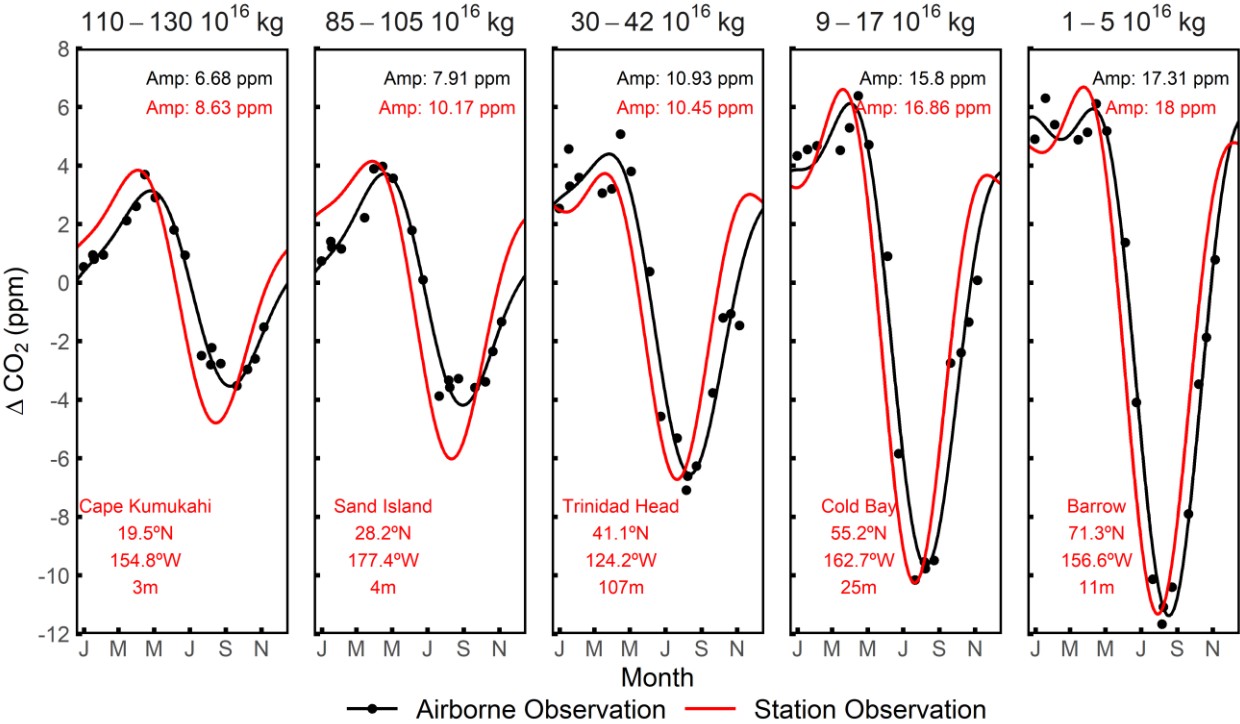

**Figure 9: CO₂ seasonal cycles of multiple surface stations (2009-2018) compared to seasonal cycles of airborne observations averaged over corresponding $M_{\theta e}$ bin. The choice of $M_{\theta e}$ bin is to approximate the range of $M_{\theta e}$ at each corresponding surface station and is shown on the top of each panel. Daily $M_{\theta e}$ of the station is computed from ERA-Interim, based on its location. We detrend station and airborne observations by subtracting the MLO stiff cubic spline trend. We compute an average detrended CO₂ for each airborne campaign transect and each $M_{\theta e}$ bin, shown as black points. The seasonal cycles are computed from a 2-harmonic fit, with the seasonal amplitude (Amp.) shown on the upper right of each panel.**






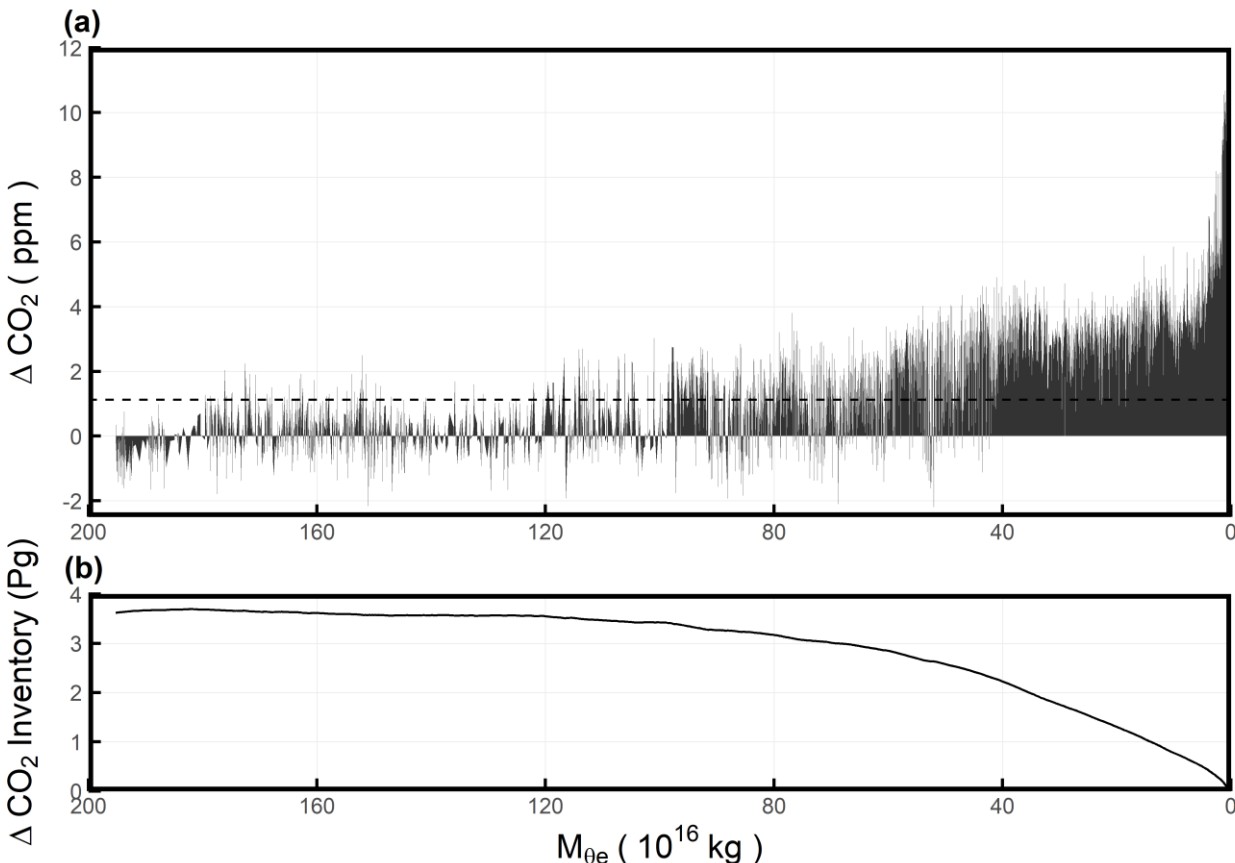

Figure 10: (a) Detrended $CO_2$ measurements from HIPPO-1 Southbound (from 12 January 2009 to 17 January 2009) plotted as a function of $M_{\theta e}$ in the Northern Hemisphere. The data are detrended by subtracting the MLO stiff cubic spline trend. Individual points are connected by straight line segments and the area under the resulting curve is shaded. We note that the area under the curve has units of ppm × kg, and dividing this by the total dry air mass (i.e., the range of $M_{\theta e}$ of the integral) gives ppm unit because the mass of dry air is proportional to the moles of dry air. The Northern Hemisphere average of 1.13 ppm is indicated by the dashed line. (b) Integral of the data in (a), rescaled from ppm to Pg, integrating from $M_{\theta e} = 0$ to a given $M_{\theta e}$ value.



**Figure 11: Comparison between the CO₂ seasonal cycle of Northern Hemisphere tropospheric average computed from airborne observation and the M$_{\theta e}$ integration method (black points and line) and the mean cycle at MLO measured by Scripps CO₂ Program from 2009 to 2018 (red line). Both are detrended by subtracting a stiff cubic spline trend at MLO. We then compute a mass-weighted average detrended CO₂ for each airborne campaign transect, shown as black points, with campaigns and transects be presented in different shapes. The seasonal cycle of both are computed by a 2-harmonic fit to the detrended time series. The 1$\sigma$ variability of the detrended average CO₂ values about the fit line is shown on the lower right. The first half year is repeated for clarity.**




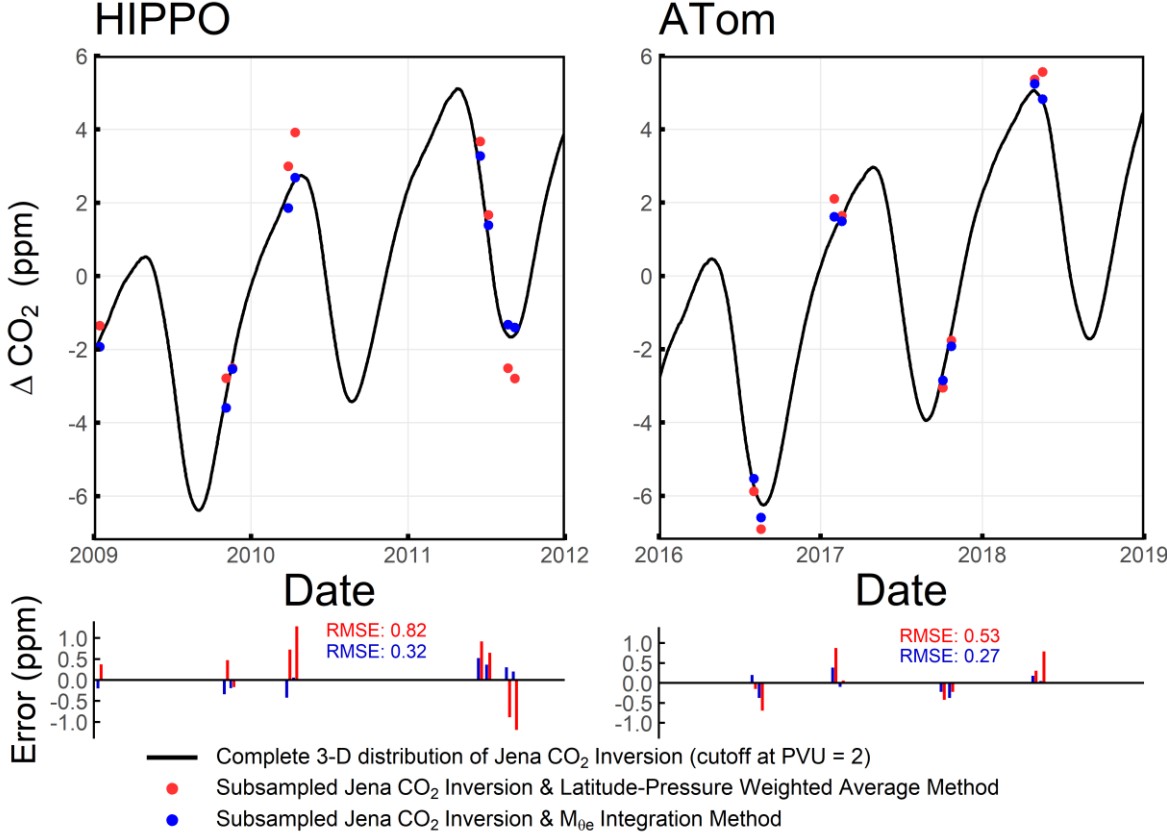

**Figure 12: Comparison between the Northern Hemisphere average CO₂ from full integration of the simulated atmospheric fields**
**from the Jena CO₂ Inversion (cutoff at PVU = 2) and from two methods that use the same simulated data subsampled with HIPPO/ATom coverage: (1) the $M_{\theta e}$ integration method (blue) and (2) simple integration by sin(latitude)-pressure (red). We divide the comparison into HIPPO (left) and ATom (right) temporal coverage. The lower panel shows the Error for individual tracks using alternate subsampling methods.**





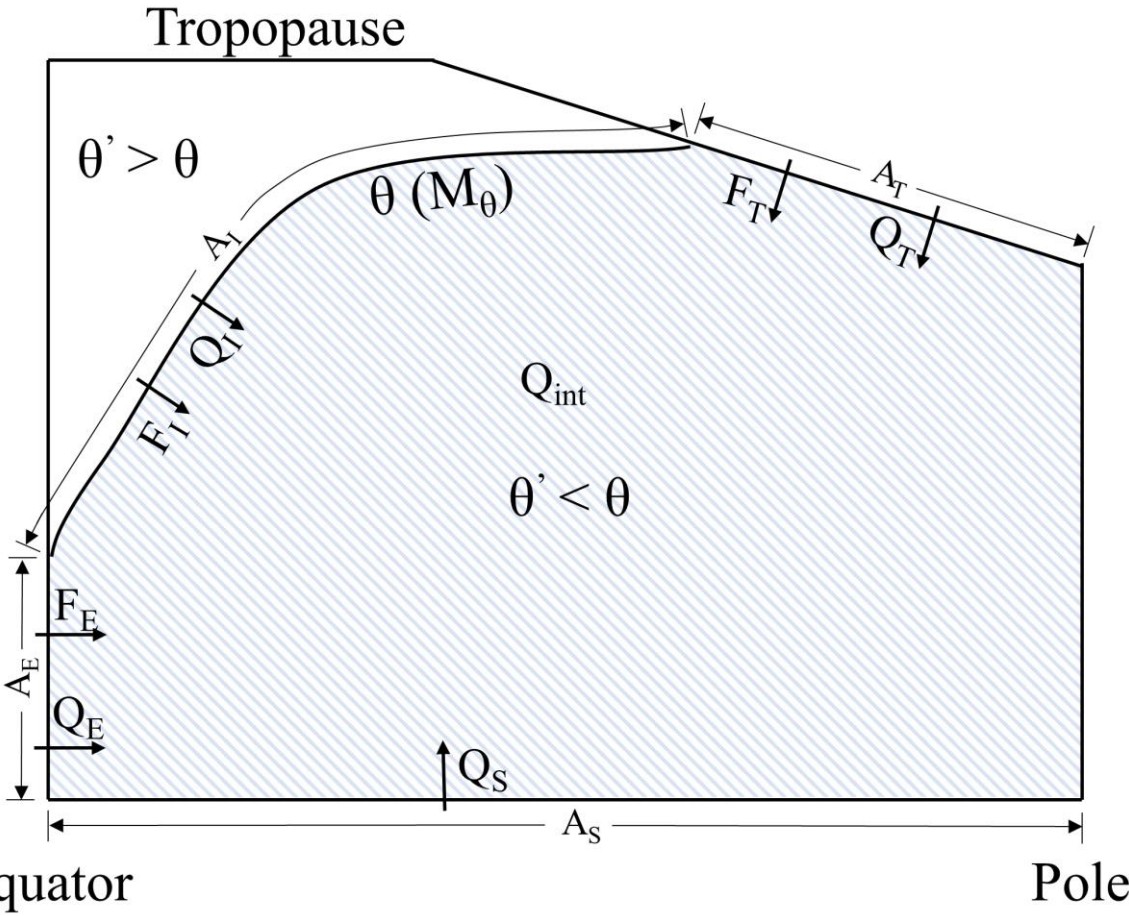


**Figure A1: Illustration of terms defined in Table A1. Shaded area denotes the region R($\theta$, t) with $\theta'$ lower than $\theta$, which is the area of mass integration to yield $M_\theta$. The curve denotes a given $\theta$ or $M_\theta$ surface.**





**Table 1: Correspondence of heating variables between our derivation (Eq. 4) and MERRA-2**

| Diabatic heating terms in our derivation (Eq. 4) | Diabatic heating terms in MERRA-2, $\frac{\partial Q_i(\theta_e,t)}{\partial \theta_e}$ |
|---|---|
| $Q'_{int}$ | 1. Radiative heating (i.e., sum of shortwave and longwave radiative heating, $Q_{rad}$) <br><br> + <br><br> 2. Energy dissipation due to dynamics computed by numeric processes ($Q_{dyn}$) <br><br> + <br><br> 3. The analysis tendency introduced during the corrector segment of the Incremental Analysis Update (IAU) cycle ($Q_{ana}$) |
| $Q_{diff} + Q_{sen}$ | 4. Turbulent heat flux including surface sensible heating ($Q_{trb}$) |
| $Q_{evap} + Q_{ice}$ | 5. Moist processes including all latent heating due to condensation and evaporation as well as the mixing by convective parameterization ($Q_{mst}$) |
| $Q_{H_2O}$ | Not available |






**Table 2: Fractional contribution of the individual heating terms in Figure 6b to their sum for $\theta_e$ = 300K. The analysis is done separately on synoptic and seasonal components. The seasonal component is based on a 2-harmonic fit and the synoptic component is defined as the residual. The fractional contributions sum to 1, while a positive contribution means in phase and negative contribution means anti-phase. A contribution in absolute value that is bigger than 1 illustrates that the variability of the heating term is larger than the variability of the sum on the corresponding time scale.**

| Heating terms | Seasonal component | Synoptic component |
|---|---|---|
| $Q_{rad}$ | 2.25 | 0.03 |
| $Q_{mst}$ | -1.39 | 0.07 |
| $Q_{dyn}$ | 0.24 | 0.72 |
| $Q_{dyn}$ | 0.21 | 0.11 |
| $Q_{ana}$ | -0.31 | 0.07 |
| Sum | 1 | 1 |



**Table 3: RMSE, seasonal amplitude and day of year of the downward zero-crossing of each simulation based on the Jena CO₂ Inversion. The true value (daily average CO₂) is computed by integrating over all tropospheric grid cells of the Jena CO₂ Inversion, while troposphere is defined by PVU < 2 from ERA-Interim. Seasonal amplitude and downward zero-crossing of true average and each simulation is computed from 2-harmonic fit to the detrended value, which is detrended by subtracting the MLO cubic stiff spline. Subsample with randomly retaining a certain fraction of data are conducted by randomly subsampling for 1000 times, thus, the seasonal amplitude and day of year of the downward zero-crossing is computed as the mean ± standard deviation of the 1000 iterations.**

| Method | RMSE (ppm)* | Seasonal Amplitude (ppm) | Downward Zero-Crossing (day) |
|---|---|---|---|
| True Value (Cut off at PVU = 2) | / | 7.58 | 175.1 |
| Evaluation of $M_{\theta e}$ Integration Method | | | |
| Full Airborne Coverage | 0.30 | 7.65 | 181.1 |
| Subsample: Equator to 30°N | 1.26 | 5.74 | 197.8 |
| Subsample: Poleward of 30°N | 0.82 | 9.47 | 179.0 |
| Subsample: Surface – 600 mbar | 0.57 | 7.77 | 185.1 |
| Subsample: 600 mbar – Tropopause | 0.38 | 7.28 | 180.7 |
| Subsample: Pacific Only | 0.33 | 7.33 | 181.6 |
| Subsample: Randomly retain 10% | 0.38 | 7.64 ± 0.116 | 182.4 ± 0.82 |
| Subsample: Randomly retain 5% | 0.40 | 7.65 ± 0.163 | 182.3 ± 1.08 |
| Subsample: Randomly retain 1% | 0.56 | 7.72 ± 0.366 | 182.2 ± 2.24 |
| Subsample: MEDUSA Coverage | 0.48 | 7.52 | 181.7 |
| Evaluation of Latitude-Pressure Weighted Average Method | | | |
| Full Airborne Coverage | 0.68 | 9.16 | 182.2 |

* Each simulation yields 17 data points of different date over the seasonal cycle from 17 airborne campaign transects. RMSE of each simulation is computed with respect to the true value.





**Table A1: Definition of variables.**

| Variable | Definition | Unit |
|---|---|---|
| $\theta'(r, t)$ | Potential temperature at location r and time t. | K |
| $\theta$ | Potential temperature of the chosen isentropic surface. | K |
| $R(\theta, t)$ | A region in which $\theta'(r, t) < \theta$ shown as shaded area in Figure A1. | |
| $A_T(\theta, t)$ | Area at the tropopause where $\theta'(r, t) < \theta$. | $m^2$ |
| $A_E(\theta, t)$ | Area at the Equator where $\theta'(r, t) < \theta$. | $m^2$ |
| $A_I(\theta, t)$ | Area where $\theta'(r, t) = \theta$. | $m^2$ |
| $A_S(\theta, t)$ | Area at the Earth surface where $\theta'(r, t) < \theta$. | $m^2$ |
| $M_\theta(\theta, t)$ | Dry air mass of $R(\theta, t)$. | kg |
| $F_T(\theta, t)$ | Mass flux through $A_T(\theta, t)$. Positive value denotes flux into region $R(\theta, t)$. | kg s$^{-1}$ |
| $F_E(\theta, t)$ | Mass flux through $A_E(\theta, t)$. Positive value denotes flux into region $R(\theta, t)$. | kg s$^{-1}$ |
| $F_I(\theta, t)$ | Mass flux through $A_I(\theta, t)$. Positive value denotes flux into region $R(\theta, t)$. | kg s$^{-1}$ |
| $Q_T(\theta, t)$ | Heat flux through $A_T(\theta, t)$. Positive value denotes flux into region $R(\theta, t)$. | J s$^{-1}$ |
| $Q_E(\theta, t)$ | Heat flux through $A_E(\theta, t)$. Positive value denotes flux into region $R(\theta, t)$. | J s$^{-1}$ |
| $Q_I(\theta, t)$ | Heat flux through $A_I(\theta, t)$. Positive value denotes flux into region $R(\theta, t)$. | J s$^{-1}$ |
| $Q_s(\theta, t)$ | Surface sensible heat flux to the region $R(\theta, t)$. Positive value denotes flux into the atmosphere. | J s$^{-1}$ |
| $Q_{int}(\theta, t)$ | Internal heating and cooling within region $R(\theta, t)$. Positive value denotes absorbing heat. | J s$^{-1}$ |
| $\dfrac{\partial Q_s(\theta, t)}{\partial \theta}$ | Surface sensible heat flux to the $\theta$ surface. Positive value denotes flux into the atmosphere (i.e., $\theta$ surface). | J s$^{-1}$ K$^{-1}$ |
| $\dfrac{\partial Q_{int}(\theta, t)}{\partial \theta}$ | Internal heating and cooling on the $\theta$ surface. Positive value denotes absorbing heat. | J s$^{-1}$ K$^{-1}$ |
| $\dfrac{\partial Q_{diff}(\theta, t)}{\partial \theta}$ | Turbulent diffusive heat fluxes into the $\theta$ surface. Positive value denotes heat flux into the $\theta$ surface | J s$^{-1}$ K$^{-1}$ |



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
