# Peer review of "A mass-weighted isentropic coordinate for mapping chemical tracers and computing atmospheric inventories"

_Atmospheric Chemistry and Physics, 2020_

## Referee Comment (RC1) · Anonymous Referee #1 · 5 Oct 2020

**Review of "A mass-weighted atmospheric isentropic coordinate for mapping chemical tracers and computing inventories" by Yuming Jin et al., submitted to ACP**

This study introduces a new isentropic coordinate based on the dry air mass below a surface of equivalent potential temperature. The goal is to better constrain the seasonal cycle of long-lived chemical species and to compute inventories of such atmospheric constituents. The first part of the manuscript is about how to calculate the dry air mass below a surface of equivalent potential temperature  $M_{\theta e}$  using reanalysis data.  $M_{\theta e}$  is then used as a new coordinate, replacing latitude, first to analyze the seasonal cycle of  $\Delta CO2$  (detrended CO2 using the time series of CO2 from the Mauna Loa Observatory) and second to calculate mass weighted averages of  $\Delta CO2$  on hemispheric scales using observational data from two recent airborne missions (Atom and HIPPO).

Traditionally, tracer distributions are viewed in geographic coordinates. However, these coordinates do not necessarily reflect the dynamics and processes which determine the tracer distributions and as such potentially add a dynamically induced variability to the tracer distribution. Alternative coordinates taking into account dynamics and/or so called transport barriers (or better said which rely on a physical rather than on a geographical basis) are a useful tool to better describe the distribution of and constrain the variability of trace species. Such coordinate transformations have for instance been used to reduce dynamically induced variability in the stratosphere (equivalent latitude instead of latitude as horizontal coordinate, e.g., Butchart and Remsberg, 1986), at the tropopause (tropopausebased instead of surface-based vertical coordinate, e.g., Birner et al., 2002), or in the Arctic (Polar Dome relative horizontal coordinate, e.g., Bozem et al., 2019). It is also the major subject of a SPARC activity. OCTAV-UTLS to reduce tracer variability in the upper troposphere and lower stratosphere (https://www.octav-utls.net/). This study adds a novel coordinate to the suite of physically based coordinates to study tracer distributions in the atmosphere and is well suited for publication in ACP. The figures are clear, the line of thought is mostly clear to me, the language is very understandable. I generally recommend publication in ACP. I mainly have several minor comments which reflect my feeling that at some points the discussion in the manuscript could provide slightly more information and which the authors might consider for a final publication.

**Comments**

- I think the introduction could provide a bit more information on the benefit of using coordinates based rather on physical than on geographical means, slightly following what is mentioned in the paragraph above.
- One question which came to my mind is about the time scales for which this coordinate may be applicable. It is stated that  $M_{\theta e}$  follows the synoptic distortions but is almost constant with respect to the seasonal cycle. So, would you then conclude that it is not applicable on the synoptic time scale? Or asked differently, is there a sort of a lower time scale limit?
- L74-77: From my point of view the mass integration which is currently in the supplement coulld be part of the main manuscript, since it is the central aspect of the first part of the manuscript. And about the mass integration, the upper boundary has only been introduced because you wanted to study the seasonal cycle of the tropospheric CO2, right?. So, let's say, if I want to study a distribution of species in the upper troposphere and lower stratosphere, the upper boundary would not be needed anymore (also surfaces of Θe become "flat" at a certain altitude and as such are an upper boundary).
- Sec 4.1: Just out of curiosity, but have you looked at CO2 in a  $\Theta_e$   $M_{\theta_e}$  coordinate system to study the seasonal cycle? Just similar to tracer distributions in an equivalent latitude-potential temperature coordinate system.
- L36ff: It is mentioned that  $M_{\theta}$  has been used for Atom and HIPPO data. Is there any reference available? Or has it only be used for internal analysis?

- L41: A bit more details at this point between the Linz study and this study would be beneficial for the reader (since potentially, not everyone interested in this tropospheric study might be familiar with the Linz study).
- L42:  $M_{\theta e}$  is mentioned here for the first time without being introduced before. This is only the case in the next paragraph.
- Sec 2.1: For each reanalysis, the number of levels is given but no more information. Could you provide at least the altitude/pressure of the top level and potentially, the level list (how much levels are roughly in the troposphere)?
- L72: It could also be mentioned here that the saturation mixing ratio of water vapor is a modified version of Wexler (1976).
- L79: Why did you not simply calculate the PV for NCEP2 data? I have no idea how well ERA-Interim and NCEP2 agree, but I wonder if it would not be better to calculate the PV for NCEP2 for consistency reasons.
- L81: Actually, what is meant with this? Do you refer to regions where pressure is not defined, such as the 850 hPa level over the Himalayan mountains?
- L90: Could you provide the range of  $\Theta_e$  values for which  $M_{\theta e}$  has been calculated?
- L94: For Figure 2, it would potentially be good to add two more panels showing the same as Fig 2a,b but not for the zonal average but for an arbitrary longitude? This would potentially help in the discussion centered around the two branches of the Hadley Circulation (L102ff). I also wonder if a  $\Theta_e$  vs  $M_{\theta e}$  plot for one or more time steps might be beneficial for the reader to get a more comprehensive idea on the relation between the two quantities and the evolution of these quantities with time.
- L96/97: This sentence confuses me. To which degree are they parallel? As stated the seasonal cycle is not similar between the two quantities. Do you mean that an  $M_{\theta e}$  surface between two  $\Theta_e$  surfaces is always parallel to these surfaces?
- L106-108: Is the displacement related to the monsoon circulations over the NH, in particular to the Asian monsoon?
- Discussion about Fig.4 : Are the fractions shown in Fig 4 constant with time?
- L122: Why do you focus on 2009? Has this year been randomly picked?
- L123: Is there a known reason why MERRA2 has this low bias? Did you check for differences in the temperature field (i.e., difference in potential temperature) and/or water content?
- Sec. 3.3: Just a note, ERA-Interim also has temperature tendencies, but I think they are only available for the forecast stream and on model levels.

- Fig. 6: Is the deviation between the blue and black curves around end of September/beginning of October a re-occurring event? Or is a random deviation for this year? How does this analysis look for other isentropes, maybe a second example could be given in the supplement? And how is the inter-annual variability?
- L177: What is meant with the term dynamic dissipation of energy?
- L191/192: Is there a reference for the NCAR UCATS and the Harvard QCL instruments?
- L193: What is meant with near-surface?
- L199: "." instead of "," after Figure 7b. Also the blue-red colorscale could be centered at 0.
- L228: Can you say something about why the airborne CO2 leads by about 10 days?
- L248: Can say something about how well CO2 is mixed on a surface of equivalent potential temperature?
- L256/260: Should the CO2 be  $\Delta$ CO2?
- L257ff: Can say something about why the inventory is dominated by this  $M_{\theta\theta}$  fraction?
- Figure 11: Please add a horizontal line at 0.
- L264: Which error are you referring to? The difference between the fit and the airborne data? Also would it make sense to add other CO2 data from the NH, eg. from Barrow to have further points for comparison.
- L338: over  $\rightarrow$  below?

**References**

Birner, T., Dörnbrack, A., and Schumann, U., How sharp is the tropopause at midlatitudes? *Geophys. Res. Lett.*, 29(14), doi:10.1029/2002GL015142, 2002.

Bozem, H., Hoor, P., Kunkel, D., Köllner, F., Schneider, J., Herber, A., Schulz, H., Leaitch, W. R., Aliabadi, A. A., Willis, M. D., Burkart, J., and Abbatt, J. P. D.: Characterization of transport regimes and the polar dome during Arctic spring and summer using in situ aircraft measurements, Atmos. Chem. Phys., 19, 15049–15071, https://doi.org/10.5194/acp-19-15049-2019, 2019.

Butchart, N., and E. E. Remsberg: The Area of the Stratospheric Polar Vortex as a Diagnostic for Tracer Transport on an Isentropic Surface. *J. Atmos. Sci.*, **43**, 1319–1339, https://doi.org/10.1175/1520-0469(1986)043<1319:TAOTSP>2.0.CO;2, 1986.

Wexler, A.: Vapor pressure formulation for water in range 0 to 100 °C. A revision. J. res. Nat. Bur. Stand., 80A, 775-785, 1976.

---

## Referee Comment (RC2) · Anonymous Referee #2 · 13 Oct 2020

This paper describes a new physically based atmospheric coordinate as an alternative to the latitude coordinate that is helpful in interpreting temporal and spatial variations of long-lived atmospheric tracers. The paper is well written, and I recommend publishing after the following minor comments are addressed.

General comment: The surfaces of $M\_theta\_e$ need to be better described. The definition of $M\_theta\_e$, given in Eq. 2, results for e.g. the northern hemisphere in a mass of a volume that encloses the north pole, reaches from the surface to the dynamical tropopause, and has as a southern boundary a specific $theta\_e$ surface. So the surface of this volume is more than just the southern boundary, and a statement

like "M_theta_e surfaces are always exactly parallel to theta_e surfaces" is difficult to understand in this context. Bins of M_theta_e in this context represent quasi differential volumes, which are more similar to surfaces of M_theta_e. This should be explained more clearly. May be a 3-D visualization of the complete surface of a given M_theta_e volume for a specific date would help illustrating this, but I'm nor sure how much distortion there would be due to synoptic disturbances that could make the volume and its surface unrecognizable.

Specific comments:

L1: The term "inventories" in the title and throughout the manuscript is a bit misleading, as it might be mistaken as e.g. emission inventories. May be the authors can use a different term such as atmospheric abundances or atmospheric burden.

L199: "Since" use lower case

References: please add a doi wherever possible (e.g. Parazoo et al., 2008 is missing the doi)
* * *

---

## Author Comment (AC1) · 30 Oct 2020

**Response to Reviewers ACP-2020-841**

We thank anonymous reviewers 1 and 2 for their valuable comments and efforts. We provide an updated manuscript, including updated figures, that incorporates the suggested changes and address specific comments below. In the following, the comments of the reviewer are presented in blue. Our responses are in black. Changes in the text are in red.

Comments from Reviewer 1

(1) I think the introduction could provide a bit more information on the benefit of using coordinates based rather on physical than on geographical means, slightly following what is mentioned in the paragraph above. We thank the reviewer for insightful suggestion and we agree that more information on the benefit together with the current application of isentropic coordinate would be helpful. We added the following sentences in the introduction:

**L30 – L42**: "A common approach to correct synoptic distortion is to use transformed coordinates rather than geographic coordinates (i.e., pressure-latitude), to take into account atmospheric dynamics and transport barriers. Such coordinate transformation has been used, for example, to reduce dynamically induced variability in the stratosphere using equivalent latitude rather than latitude as horizontal coordinate (Butchart and Remsberg, 1986), to diagnose tropopause profile using tropopause-based rather than surface-based vertical coordinate (Birner et al., 2002), to study transport regime in the Arctic using a horizontal coordinate based on Polar Dome (Bozem et al., 2019), and to study UTLS (Upper Troposphere Lower Stratosphere) tracer data by using tropopause-based, jet-based, and equivalent latitude coordinates (Irina et al., 2019). In the troposphere, a transformed coordinate, isentropic coordinate ($\theta$) has been widely applied to evaluate the distribution of tracer data (Miyazaki et al., 2008; Parazoo et al., 2011, 2012). As air parcels move with synoptic disturbances, $\theta$ and the tracer tend to be similarly displaced so that the $\theta$-tracer relationship is relatively conserved (Keppel-Aleks et al., 2011). Furthermore, vertical mixing tends to be rapid on $\theta$ surfaces, so $\theta$ and tracer contours are often nearly parallel (Barnes et al., 2016). However, $\theta$ varies greatly with latitude and altitude over seasons due to changes in heating and cooling with solar insolation, which complicates the interpretation of $\theta$-tracer relationships on seasonal time scales."

(2) One question which came to my mind is about the time scales for which this coordinate may be applicable. It is stated that $M_{\theta e}$ follows the synoptic distortions but is almost constant with respect to the seasonal cycle. So, would you then conclude that it is not applicable on the synoptic time scale? Or asked differently, is there a sort of a lower time scale limit?

$M_{\theta e}$ moves in parallel with the corresponding $\theta_e$ surface on the synoptic time scale, which is useful in correcting this dynamic-induced disturbances. When applied to sparse airborne observations, $M_{\theta e}$ is similar to $\theta_e$ in correcting these disturbances on the short time scale. However, $\theta_e$ surfaces move meridionally with season, which is not suitable to study the seasonal cycle, since a given $\theta_e$ surface represents different part of the atmosphere in different seasons. Under this circumstance, $M_{\theta e}$, which has less seasonality, is useful to map a tracer. The advantage of $M_{\theta e}$ is that, on the synoptic time scale, it corrects the synoptic disturbances, like $\theta_{e,}$ to compute a mean tracer concentration with high accuracy, and on the seasonal scale, represents a similar part of the atmosphere, to be useful in mapping the seasonal cycle of a tracer. Besides, on the synoptic time scales, $M_{\theta e}$ may have an advantage over $\theta_e$ in making it easier to compute atmospheric inventories.

(3) L74-77: From my point of view the mass integration which is currently in the supplement could be part of the main manuscript, since it is the central aspect of the first part of the manuscript. And about the mass integration, the upper boundary has only been introduced because you wanted to study the seasonal cycle of the tropospheric $CO_2$, right? So, let's say, if I want to study a distribution of species in the upper troposphere and lower stratosphere, the upper boundary would not be needed anymore (also surfaces of $\theta_e$ become "flat" at a certain altitude and as such are an upper boundary).

We moved Supplementary S1 (i.e., method of mass integration) into Section 2.2 (**L81 − L94**). We have upper boundary at tropopause because we are only interested in $CO_2$ distribution in the troposphere for our application. For other applications, for example, tracer distribution in the upper troposphere and lower stratosphere, the upper boundary is not necessary or should be set at the top of the stratosphere. We concluded in our discussion (**L425**) that the set of boundary conditions should be based on target of interest.

(4)Sec 4.1: Just out of curiosity, but have you looked at CO2 in a Θe - Mθe coordinate system to study the seasonal cycle? Just similar to tracer distributions in an equivalent latitude-potential temperature coordinate system.

We haven't looked at it, but this is a good suggestion for future analysis.

(5)L36ff: It is mentioned that Mθ has been used for Atom and HIPPO data. Is there any reference available? Or has it only be used for internal analysis?

The idea of $M_{\theta e}$ stems from analysis of ATom and HIPPO data so it has only been used for internal analysis.

(6) L41: A bit more details at this point between the Linz study and this study would be beneficial for the reader (since potentially, not everyone interested in this tropospheric study might be familiar with the Linz study).

We thank the reviewer for the valuable suggestion. We added the following sentence to better describe Linz study:

**L49 – L50**: "We note that a similar concept to $M_{\theta e}$ has been introduced in the stratosphere by Linz et al. (2016), in which $M(\theta)$ is defined as the mass above the $\theta$ surface, to study the relationship between age of air and diabatic circulation of the stratosphere."

(7) L42: $M_{\theta e}$ is mentioned here for the first time without being introduced before. This is only the case in the next paragraph.

The first time we mention $M_{\theta e}$ is in the sentence "we have found it useful to transform potential temperature into a mass-based unit, $M_\theta$, which we define as the total mass of dry air under a given isentropic surface in the hemisphere." This sentence occurs before line 42.

(8) Sec 2.1: For each reanalysis, the number of levels is given but no more information. Could you provide at least the altitude/pressure of the top level and potentially, the level list (how much levels are roughly in the troposphere)?

We thank the reviewer for the suggestion. We added the following sentences to describe the vertical levels of each reanalysis product in more detail.

**L69 – L74**: "All products have 2.5° horizontal resolution. NCEP2 has daily resolution and we average 6-hourly ERA-Interim fields and 3-hourly MERRA2 fields to yield daily fields. ERA-Interim has 32 vertical levels from 1000 mbar to 1 mbar, with approximately 20 to 27 levels in the troposphere. NCEP2 has 17 vertical levels from 1000 mbar to 10 mbar, with approximately 8 to 12 levels in the troposphere. MERRA2 has 42 vertical levels from 985 mbar to 0.01 mbar, with approximately 21 to 25 levels in the troposphere."

(9) L72: It could also be mentioned here that the saturation mixing ratio of water vapor is a modified version of Wexler (1976).

Thanks for pointing it out. We now mention this in **L81**: "Following Bolton (1980), we compute water vapor mixing ratio (w) from relative humidity (RH, kg kg$^{-1}$) provided by the reanalysis products and the formula for saturation mixing ratio of water vapor ($P_{s,v}$, mbar) modified by Wexler (1976)."

(10) L79: Why did you not simply calculate the PV for NCEP2 data? I have no idea how well ERA- Interim and NCEP2 agree, but I wonder if it would not be better to calculate the PV for NCEP2 for consistency reasons.

NCEP2 potential vorticity is not directly available at pressure level, so we interpolated ERA-Interim PV to the NCEP2 fields. We have calculated PV for NCEP2 and applied the new upper boundary for NCEP2 mass integration.

(11) L81: Actually, what is meant with this? Do you refer to regions where pressure is not defined, such as the 850 hPa level over the Himalayan mountains?

Yes. We added the following sentence to make this description more clear.

L99-100: "ERA-Interim and NCEP2 include hypothetical levels below the true land/sea surface, for example, the 850 hPa level over the Himalayan, which we exclude in the calculation of $M_{\theta e}$."

(12) L90: Could you provide the range of $\Theta e$ values for which $M\theta e$ has been calculated?

 The $\theta_e$-$M_{\theta e}$ lookup table provided spans from the lowest to the highest $\theta_e$ surface in the troposphere of each day. Therefore the range of $\theta_e$ of $M_{\theta e}$ is different day by day but contains all value in the troposphere. We added the following sentence to better describe the $\theta_e$ range.

**L108 – L110**: "We provide this look-up table for each hemisphere computed from ERA-Interim from 1980 to 2018 with daily resolution and from the lowest to the highest $\theta_e$ surface in the troposphere with 1 K interval (see data availability)."

(13) L94: For Figure 2, it would potentially be good to add two more panels showing the same as Fig 2a,b but not for the zonal average but for an arbitrary longitude? This would potentially help in the discussion centered around the two branches of the Hadley Circulation (L102ff). I also wonder if a $\Theta e$ vs $M\theta e$ plot for one or more time steps might be beneficial for the reader to get a more comprehensive idea on the relation between the two quantities and the evolution of these quantities with time.

We thank the reviewer for the suggestion. We added another two panels in new Figure 3, showing Jan-2009 average and July-2009 average $M_{\theta e}$ cross sections at 180E and 100E. This figure also clearly shows the difference in summer to winter $M_{\theta e}$ displacements over the ocean and land. We find that adding more time steps, by adding more line types, only makes the figure too noisy to visualize. We think summer and winter $M_{\theta e}$ cross section alone could help to sharply focus on the seasonal variability of $M_{\theta e}$, without being distracted with too many details.  For the new panels, we added more description of $M_{\theta e}$ displacements as the zonal average and at these two meridians as following.

**L123 – L141:"** $M_{\theta e}$ surfaces at given meridians (Figure 3) in the Northern Hemisphere show clear zonal asymmetry, with larger and more complex displacements compared to the zonal averages, associated with differential heating by land and ocean, and orographic stationary Rossby waves (Hoskins and Karoly, 1981; Wills and Schneider, 2018).  For example, over the Northern Hemisphere ocean at 180°E (Figure 3a) and from the summer to winter, $M_{\theta e}$ surfaces move poleward in the mid- to high latitude (e.g. poleward of 45°N), but move equatorward in the mid- to low latitude lower troposphere (e.g. equatorward of 45°N, 900 – 700 mbar), with the magnitude smaller than 10 degrees latitude in both. Whereas, over the Northern Hemisphere land at 100°E (Figure 3b) and from the summer to winter, $M_{\theta e}$ surfaces moves equatorward by up to 30 degrees latitude, except high latitude middle troposphere (e.g. poleward of 70°N, ~ 500 mbar), where the flat $M_{\theta e}$ surfaces lead to slightly poleward displacements. In the Southern Hemisphere, in contrast, the summer to winter displacements of the 180°E and 100°E sections are similar to the zonal average.

At lower latitudes, the zonal averages of $M_{\theta e}$ and $\theta_e$ both exhibit strong secondary maxima near the surface associated with the Hadley circulation (Equatorward of 30° N/S) and in the summer, driven by high water vapor. From the contours in Figure 2, this surface branch of high $M_{\theta e}$ and $\theta_e$ appears disconnected from the upper tropospheric branch. In fact, these two branches are connected through air columns undergoing deep convection, which are not resolved in the zonal means shown in Figure 2, but are resolved in some meridians (e.g. Figure 3a). We also note that, over the land at 100°E (Figure 3b), the two disconnected $M_{\theta e}$ and $\theta_e$ branches in the Northern Hemisphere summer are displaced poleward compared to the zonal average, consistent with a northward shift of intertropical convergence zone (ITCZ) over southern Asia. The existence of these two branches may limit some applications of $M_{\theta e}$, as discussed in Section 4."

[Figure]

Figure 3: (a) $M_{\theta e}$ surfaces at 180°E as Jan-2009 average (solid lines) and July-2009 average (dashed lines). This cross section is mostly over the Pacific Ocean. (b) $M_{\theta e}$ surfaces at 100°E as Jan-2009 average (solid lines) and July-2009 average (dashed lines). This cross section is mostly over the Eurasia land in the Northern Hemisphere. $M_{\theta e}$ and tropopause are computed from ERA-Interim.

(14) L96/97: This sentence confuses me. To which degree are they parallel? As stated the seasonal cycle is not similar between the two quantities. Do you mean that an Mθe surface between two Θe surfaces is always parallel to these surfaces?

Given a specific $\theta_e$ surface, the $M_{\theta e}$ surface with the given $\theta_e$ value is parallel to the $\theta_e$ surface geographically, since the way we compute the $M_{\theta e}$ surface is summing all air mass with a lower equivalent potential temperature to the given $\theta_e$ surface. In other words, on a given $\theta_e$ surface, the $M_{\theta e}$ value has to be the same. To avoid any misleading, we updated the sentence as following.

**L114 – L116:** "By definition, each $M_{\theta e}$ surface is exactly aligned with a corresponding $\theta_e$ surface, and $M_{\theta e}$ surfaces have the same characteristics as $\theta_e$ surfaces, which decrease with latitude and generally increase with altitude."

(15) L106-108: Is the displacement related to the monsoon circulations over the NH, in particular to the Asian monsoon?

We suspect that this tilting and displacement could be explained by different energy fluxes between land and the ocean, which might be related to the monsoon circulation. From the updated Figure 3, it is also clear that the summer to winter displacement of $M_{\theta e}$ surfaces are different between land and ocean.

(16) Discussion about Fig.4 : Are the fractions shown in Fig 4 constant with time?

The mass fraction is almost constant with time with slightly seasonal variation in the lower troposphere and high or low $M_{\theta e}$ bands. We added the following sentence for discussion.

**L151 – L154:** "The contribution from the surface to 800 mbar increases as $M_{\theta e}$ increases above 120 ($10^{16}$ kg). The mass fraction shows only small variations with season, with the lower troposphere (Surface to 800 mbar) contributing slightly less in the low $M_{\theta e}$ bands and slightly more in the high $M_{\theta e}$ bands in the summer, which is closely related to the seasonal tilting of corresponding $\theta_e$ surfaces."

(17) L122: Why do you focus on 2009? Has this year been randomly picked?

The choice of 2009 is arbitrary, and mostly because the first year of HIPPO campaign is in the year of 2009. We edited the following:

**L113:** "Figure 2 shows snapshots of the distribution of zonal average $\theta_e$ and $M_{\theta e}$ with latitude and pressure at two arbitrary time slices (1 January 2009, 1 July 2009)"

(18) L123: Is there a known reason why MERRA2 has this low bias? Did you check for differences in the temperature field (i.e., difference in potential temperature) and/or water content?

We use $M_{\theta e}$ computed from ERA-Interim as a target to compare with $M_{\theta e}$ computed from MERRA2 and NCEP2. Among these two, MERRA2 shows a smaller difference regarding to ERA-Interim. This does not mean MERRA-2 has a low bias but shows that the difference between MERRA2 and ERA-Interim is smaller than the difference between NCEP2 and ERA-Interim. We do not well understand the reason that MERRA2 shows a smaller difference, but we suspect that it is related to the methodology of different reanalysis products.

We also checked the difference in temperature field at the same longitude, latitude and pressure between three reanalysis products. ERA-Interim and MERRA2 generally shows the smallest difference. The standard deviation of this differences in the year of 2009 is 0.72 K between ERA-Interim and MERRA2, which is about half of that (1.47K) between ERA-Interim and NCEP2.

(19) Sec. 3.3: Just a note, ERA-Interim also has temperature tendencies, but I think they are only available for the forecast stream and on model levels.

We thank the reviewer for pointing this out. We need temperature tendencies in the reanalysis rather than forecast.

(20) Fig. 6: Is the deviation between the blue and black curves around end of September/beginning of October a re-occurring event? Or is a random deviation for this year? How does this analysis look for other isentropes, maybe a second example could be given in the supplement? And how is the inter-annual variability?

We thank the reviewer for this interesting question. In response, we repeated the analysis on the 300 K $\theta_e$ surfaces for another five years (2010-2011, and 2016-2018) and found that this is indeed a re-occurring feature. This feature also occurs on some lower surfaces (e.g. 290 K) but not higher surfaces. We hypothesize that this event is potentially due to the assumption we made $Q_{mst}$ approximates the sum of $Q_{ice}$ and $Q_{evap}$, and/or the underestimation of cooling of the radiative term or moisture term in the mid- to high latitude in MERRA2. We note that an underestimation of cooling would underestimate the equatorward shift of the $M_{\theta e}$ surface, which leads to the underestimation of $\frac{dM_{\theta_e}}{dt}$. We now provide further discussion about this feature, and include results for different years and on different moist isentropes in the supplement:

**L210 – L214**: "Figure 7a shows poorer agreement from late August to October, which we also find in other years (Figure S1 and S2), and on lower (e.g., $\theta_e = 290K$, Figure S3) but not higher surfaces (e.g., $\theta_e = 310K$, Figure S4), where the two methods agree better. The poor agreement may reflect a partial breakdown of the assumption that $Q_{mst}$ approximates the sum of $Q_{ice}$ and $Q_{evap}$, but further analysis is beyond the scope of this study."

[Figure]

Figure S1: (a) Temporal variation of $M_{\theta e}$ in the Northern Hemisphere at $\theta_e$ = 300 K computed by integrating air mass (blue line) and estimated from the sum of five heating terms (Table 1) in MERRA-2 (black line). (b) The heating variables decomposed into five contributions as indicated (see Table 1). Results shown are for year 2010.

[Figure]

Figure S2: Similar to Figure S1, but for the year of 2011.

[Figure]

Figure S3: Similar to Figure S1, but for the year of 2009 and on the 290K $\theta_e$ surface.

[Figure]

Figure S4: Similar to Figure S3, but on the 310K θ_e surface.

(21) L177: What is meant with the term dynamic dissipation of energy?

Dynamic dissipation of energy means the dissipation of the kinetic energy of turbulence (the energy associated with turbulent eddies in a fluid flow). This is the rate at which the turbulence energy is absorbed by breaking the eddies down into smaller and smaller eddies until it is ultimately converted into heat by viscous forces. To avoid any misleading of the word, we replace all 'Dynamic Dissipation of Energy' with 'Dissipation of Kinetic Energy of Turbulence'. This term comes from 'MERRA-2: File Specification' (https://gmao.gsfc.nasa.gov/pubs/docs/Bosilovich785.pdf).

(22) L191/192: Is there a reference for the NCAR UCATS and the Harvard QCL instruments?

We added reference for the NCAR UCATS and Harvard QCLS instruments.

(23) L193: What is meant with near-surface?

We filtered out all airborne observations within ~ 100 seconds since the take off and within ~ 600 seconds before the landing. This filter is decided manually to discard observation potentially polluted during the take off and landing of the aircraft. We expanded the following sentence:

**L230 - 231:** "Furthermore, we exclude all near-surface observations within ~ 100 seconds of take-offs, within ~ 600 seconds of landings, and missed approaches, which usually show high $CO_2$ variability due to strong local influences."

(24) L199: "." instead of "," after Figure 7b. Also the blue-red colorscale could be centered at 0.

Thanks. We changed as suggested. The updated figure is shown as following.

[Figure]

Figure 8: (a) HIPPO and ATom horizontal flight tracks coloured by campaigns. (b) Latitude and pressure cross-section of detrended $CO_2$ of each airborne campaign transect. $CO_2$ is detrended by subtracting MLO stiff cubic spline trend, which is computed by a stiff cubic spline function plus 4-harmonic functions with linear gain to MLO record.

(25) L228: Can you say something about why the airborne CO2 leads by about 10 days?

This comparison uses airborne data from a wide range of $M_{\theta e}$ (90 – 110 $10^{16}$ kg) and pressure (500 – 800, mbar), while $M_\theta$ and pressure at MLO is about 90 - 100 $10^{16}$ kg and ~ 670 mbar. Therefore, we expect this small difference in amplitude and phase and this difference is within the 1σ uncertainty of our estimation of airborne observation. We added following sentences to discuss the difference.

**L266 – L268**: "This small difference is within the 1σ uncertainty of our estimation from airborne observation, and some difference is expected, since we choose a $M_{\theta e}$-pressure bin wider than the seasonal variation of $M_{\theta e}$ and pressure at MLO."

(26) L248: Can say something about how well $CO_2$ is mixed on a surface of equivalent potential temperature? Barnes et al. 2016 shows that $\theta$ contours and $CO_2$ contours are nearly parallel. Thus, much of synoptic transport is along surfaces of constant potential temperatures. Parazoo et al., 2011 & 2012 and Miyazaki et al., 2008 shows that $CO_2$ tends to transport fast along moist isentropes especially along the mid-latitude storm track. Our analysis in Figure 13 based on Jena simulation also demonstrates that this assumption is reasonably well.

(27) L256/260: Should the CO2 be ΔCO2?
Thanks. We changed as suggested.

(28) L257ff: Can say something about why the inventory is dominated by this MΘe fraction?
The inventory here is computed based on the detrended $CO_2$, therefore, ΔInventory is largely depended on the seasonal cycle of $CO_2$. In the low latitude (high $M_{\theta e}$), $CO_2$ doesn't deviate as much from its annual mean. In the  mid- to high latitude (low $M_{\theta e}$), $CO_2$ has large seasonal cycle driven by temperate and boreal forests. Therefore, $\Delta CO_2$ inventory is dominated by the high $M_{\theta e}$ bands (mid- to high latitude). We added the following sentence.

**L299 – L302:** "The $\Delta CO_2$ atmospheric inventory is dominated by the domain $M_{\theta e} < 120$ (mid- to high latitude), which has large $CO_2$ seasonal cycle driven by temperate and boreal ecosystem, with less than 4.1% contributed by the additional ~38.8% of the air mass outside this domain in the low latitude or upper troposphere (Fig. 11b), where $\Delta CO_2$ differs less from the detrended baseline, which has been subtracted."

(29) Figure 11: Please add a horizontal line at 0.
Thanks. We added a light grey grid, for better visualization:

[Figure]

Figure 12: Comparison between the CO$_2$ seasonal cycle of Northern Hemisphere tropospheric average computed from airborne observation and the M$_{\theta e}$ integration method (black points and line) and the mean cycle at MLO measured by Scripps CO$_2$ Program from 2009 to 2018 (red line). Both are detrended by subtracting a stiff cubic spline trend at MLO. We then compute a mass-weighted average detrended CO$_2$ for each airborne campaign transect, shown as black points, with campaigns and transects be presented in different shapes. The seasonal cycle of both are computed by a 2-harmonic fit to the detrended time series. The 1$\sigma$ variability of the detrended average CO$_2$ values about the fit line is shown on the lower right. The first half year is repeated for clarity.

(30) L264: Which error are you referring to? The difference between the fit and the airborne data? Also would it make sense to add other CO2 data from the NH, eg. from Barrow to have further points for comparison.

We are referring to the error of our computed Northern Hemisphere mass-weighted average CO$_2$ seasonal cycle. This is the deviation between our estimation from airborne data and the true average. This error comes from the limited coverage of airborne data in space and time as well as measurement irreproducibility. Since the true average is unknown, we have to simulate a true average in order to estimate the bias. We utilize the atmospheric CO$_2$ fields from the Jena CO$_2$ Inversion for this simulation. Since MLO is often viewed as the Northern Hemisphere average, we show the comparison of our estimation and MLO seasonal cycle only to focus on their differences. This comparison is not part of the error estimation. To make this description more clear, we edited the following sentence.

**L307 – 308:** "To address the error in our estimation of Northern Hemisphere mass-weighted average $CO_2$ seasonal cycle from HIPPO and Atom airborne observation,"

(31) L338: over → below?

Thanks. We changed as suggested.

Comments from reviewer 2

(1) The surfaces of M_theta_e need to be better described. The definition of M_theta_e, given in Eq. 2, results for e.g. the northern hemisphere in a mass of a volume that encloses the north pole, reaches from the surface to the dynamical tropopause, and has as a southern boundary a specific theta_e surface. So the surface of this volume is more than just the southern boundary, and a statement like "M_theta_e surfaces are always exactly parallel to theta_e surfaces" is difficult to understand in this context. Bins of M_theta_e in this context represent quasi differential volumes, which are more similar to surfaces of M_theta_e. This should be explained more clearly. May be a 3-D visualization of the complete surface of a given M_theta_e volume for a specific date would help illustrating this, but I'm nor sure how much distortion there would be due to synoptic disturbances that could make the volume and its surface unrecognizable.

We thank the reviewer for the suggestion. This comment is similar to comment 13 from the first reviewer, in which we've provided a detailed reply. We want to reiterate that, $M_{\theta e}$ labels the entire volume, but the corresponding $M_{\theta e}$ surface represents the upper surface for mass integration. Also, any $M_{\theta e}$ surface might intercept the Equator and tropopause. Given the $\theta_e$ surface, the $M_{\theta e}$ surface is parallel to the $\theta_e$ surface with the same $\theta_e$ value, therefore, any $M_{\theta e}$ surface is parallel to the corresponding $\theta_e$ surface. We think a 3-D plot would be too noisy to visualize. Alternately, we add a new Figure 3 to show Jan-2009 average and July-2009 average $M_{\theta e}$ cross sections at 180E (over the ocean in the Northern Hemisphere) and 100E (over the land in the Northern Hemisphere). For more information, one can refer to our reply to comment 13 of the first reviewer.

(2) L1: The term "inventories" in the title and throughout the manuscript is a bit misleading, as it might be mistaken as e.g. emission inventories. May be the authors can use a different term such as atmospheric abundances or atmospheric burden.

We thank the reviewer for the suggestion. We would use 'atmospheric inventory' rather than 'inventory' in the title and manuscript. We think this change would make it clear that we are talking about abundances of atmospheric tracer rather than emission inventories.

(3) L199: "Since" use lower case

The ',' before 'Since' should be '.'. We have fixed it.

(4) please add a doi wherever possible (e.g. Parazoo et al., 2008 is missing the doi)

We thank the reviewer for pointing it out. We have fixed it.

[revised manuscript text omitted]

30    Figure S3: Similar to Figure S1, but for the year of 2009 and on the 290 K θ$_e$ surface.

[Figure]

Figure S4: Similar to Figure S3, but on the 310 K θ$_e$ surface.

**Table S1: Number of data points of each airborne campaign transect for each simulation**

| Airborne Transect | Original | Equator to 30 °N | Poleward of 30 °N | Surface – 600 mbar | 600 mbar – Trop. | Pacific Only | Medusa Coverage | Random 10 % | Random 5 % | Random 1 % |
|---|---|---|---|---|---|---|---|---|---|---|
| HIPPO1 SB | 4837 | 1454 | 3383 | 1794 | 3043 | 4837 | 76 | 484 | 242 | 48 |
| HIPPO2 SB | 4665 | 1510 | 3155 | 1945 | 2720 | 4665 | 82 | 451 | 233 | 45 |
| HIPPO2 NB | 5508 | 2428 | 3080 | 2159 | 3349 | 5508 | 93 | 543 | 275 | 54 |
| HIPPO3 SB | 4439 | 1371 | 3068 | 2038 | 2401 | 4439 | 88 | 427 | 222 | 43 |
| HIPPO3 NB | 4086 | 1135 | 2951 | 1790 | 2296 | 4086 | 84 | 399 | 204 | 40 |
| HIPPO4 SB | 5491 | 1602 | 3889 | 2340 | 3151 | 5491 | 81 | 534 | 275 | 53 |
| HIPPO4 NB | 6411 | 3134 | 3277 | 3142 | 3269 | 6411 | 124 | 626 | 321 | 63 |
| HIPPO5 SB | 5538 | 1678 | 3860 | 2569 | 2969 | 5538 | 78 | 548 | 277 | 55 |
| HIPPO5 NB | 4715 | 1705 | 3010 | 2066 | 2649 | 4715 | 86 | 392 | 236 | 39 |
| ATom1 SB | 9832 | 2333 | 7499 | 3186 | 6646 | 9832 | 83 | 455 | 492 | 46 |
| ATom1 NB | 10685 | 3186 | 7499 | 3665 | 7020 | 0 | 59 | 893 | 534 | 89 |
| ATom2 SB | 11372 | 3909 | 7463 | 4057 | 7315 | 11372 | 84 | 1109 | 569 | 111 |
| ATom2 NB | 10741 | 3284 | 7457 | 3792 | 6949 | 0 | 91 | 1042 | 537 | 104 |
| ATom3 SB | 15143 | 3751 | 11392 | 4817 | 10326 | 15143 | 87 | 1460 | 757 | 146 |
| ATom3 NB | 14039 | 4173 | 9866 | 4764 | 9275 | 0 | 92 | 1362 | 702 | 136 |
| ATom4 SB | 13554 | 3683 | 9871 | 5249 | 8305 | 13554 | 84 | 1327 | 678 | 132 |
| ATom4 NB | 11995 | 3626 | 8369 | 4130 | 7865 | 0 | 89 | 1187 | 600 | 119 |

**References**

Graven, H. D., Keeling, R. F., Piper, S. C., Patra, P. K., Stephens, B. B., Wofsy, S. C., Welp, L. R., Sweeney, C., Tans., P. P., Kelley, J. J., Daube, B. C., Kort, E. A., Santoni, G. W. and Bent, J. D.: Enhanced seasonal exchange of $CO_2$ by Northern ecosystems since 1960, Science, 341(6150), 1085–1089, doi:10.1126/science.1239207, 2013.